# Wave patterns organize cellular protrusions and control cortical dynamics

Yuchuan Miao[1,2,†] ⓘD, Sayak Bhattacharya[3,†] ⓘD, Tatsat Banerjee[2,4], Bedri Abubaker-Sharif[2], Yu Long[2], Takanari Inoue[2], Pablo A Iglesias[2,3,*] ⓘD & Peter N Devreotes[2,**] ⓘD

## Abstract

Cellular protrusions are typically considered as distinct structures associated with specific regulators. However, we found that these regulators coordinately localize as propagating cortical waves, suggesting a common underlying mechanism. These molecular events fell into two excitable networks, the signal transduction network STEN and the cytoskeletal network CEN with different wave substructures. Computational studies using a coupled-network model reproduced these features and showed that the morphology and kinetics of the waves depended on strengths of feedback loops. Chemically induced dimerization at multiple nodes produced distinct, coordinated alterations in patterns of other network components. Taken together, these studies indicate: STEN positive feedback is mediated by mutual inhibition between Ras/Rap and PIP2, while negative feedback depends on delayed PKB activation; PKBs link STEN to CEN; CEN includes positive feedback between Rac and F-actin, and exerts fast positive and slow negative feedbacks to STEN. The alterations produced protrusions resembling filopodia, ruffles, pseudopodia, or lamellipodia, suggesting that these structures arise from a common regulatory mechanism and that the overall state of the STEN-CEN system determines cellular morphology.

**Keywords** cell migration; cellular protrusion; complex network; excitable system; pattern formation

**Subject Categories** Cell Adhesion, Polarity & Cytoskeleton; Quantitative Biology & Dynamical Systems; Signal Transduction

**Mol Syst Biol. (2019) 15: e8585**

## Introduction

Though there has been great interest since Alan Turing's research (Turing, 1952) in understanding the molecular basis for generating patterns in biological organisms, it is only recently that wave patterns in individual cells and their effect on cellular morphology and function have been considered. Waves of cytoskeletal and signaling events have been observed at the cortex of various cell types including *Dictyostelium* (Vicker, 2002; Bretschneider *et al*, 2009; Arai *et al*, 2010), neutrophils (Weiner *et al*, 2007), fibroblasts (Case & Waterman, 2011), *Xenopus* eggs (Bement *et al*, 2015), mast cells (Wu *et al*, 2013; Xiong *et al*, 2016), developing neurons (Winans *et al*, 2016), and cancer cells (Marchesin *et al*, 2015). Increasing evidence suggests that these wave patterns underlie fundamental processes such as cell division, migration, phagocytosis, polarization, and gene expression. In mast cells, waves are proposed to coordinate release of granules (Yang *et al*, 2017), while in developing neurons, waves are suggested to widen axons to facilitate cargo transport (Winans *et al*, 2016). In migratory cells, waves are linked with protrusions that drive cell motility (Weiner *et al*, 2007; Huang *et al*, 2013; Devreotes *et al*, 2017) and it has been shown that changing wave properties can significantly alter migratory modes of *Dictyostelium* cells (Miao *et al*, 2017). Although there is rich theory regarding wave characteristics (reviewed in Tyson & Keener, 1988), how this connects with the underlying molecular architecture driving cortical waves is largely unknown.

Mathematical models consisting of reaction–diffusion equations describing an activator–inhibitor excitable system have been able to capture wave formation and propagation in biological systems realistically (Meinhardt & de Boer, 2001; Hecht *et al*, 2010; Xiong *et al*, 2010). In these models, the activator triggers a fast, autocatalytic loop that generates positive feedback, and a slower inhibitor that forms a negative feedback loop. As the activator and inhibitor diffuse, the activity of these processes spreads throughout the medium in the form of propagating wave. Cytoskeletal and signaling events involved in waves display all-or-nothing responses, refractory periods, and annihilation upon wave collision—all hallmarks of excitable systems—supporting the merits of these excitability-based models (Huang *et al*, 2013). These non-linear feedback models suggest that interactions of multiple components are necessary to generate excitable waves, and that alteration in any component can have a significant impact on the overall network. By altering the

1    Department of Biological Chemistry, School of Medicine, Johns Hopkins University, Baltimore, MD, USA
2    Department of Cell Biology and Center for Cell Dynamics, School of Medicine, Johns Hopkins University, Baltimore, MD, USA
3    Department of Electrical and Computer Engineering, Whiting School of Engineering, Johns Hopkins University, Baltimore, MD, USA
4    Department of Chemical and Biomolecular Engineering, Whiting School of Engineering, Johns Hopkins University, Baltimore, MD, USA
    *Corresponding author. Tel: +1 410 516 6026; E-mail: pi@jhu.edu
    **Corresponding author. Tel: +1 410 955 3225; E-mail: pnd@jhmi.edu
    † These authors contributed equally to this work

strengths of various loops in these models and comparing the ensuing patterns with experimental observations, the nature of the underlying biochemical network can be elucidated.

The complexity of molecular events and interactions poses great challenge for in-depth understanding of wave formation and cellular protrusions at the molecular level. In multiple cell lines, numerous molecules have been identified that are present in or are associated with cortical waves. In *Dictyostelium*, components including F-actin, phosphoinositides, small GTPases, kinases, and phosphatases, have been shown to participate in wave propagation (Gerisch *et al*, 2011; Devreotes *et al*, 2017). The numerous components are spatially and temporally coordinated with each other; thus, localization characterization alone is insufficient to uncover the molecular network. Component-specific perturbations are needed to reveal molecular interactions at each node. In addition, these perturbations have to be abrupt, in the time scale of seconds to minutes, to match the highly dynamic nature of cortical waves (Gerhardt *et al*, 2014; Miao *et al*, 2017). Here, we use acute, targeted perturbations, analyze their effect on wave organization, and compare these observations with theoretical predictions to map out the molecular connections underlying the dynamic patterns. Further, we show a close correlation between different cortical patterns and various cellular protrusions involved in cell migration. In contrast to the traditional view that diverse cellular protrusions are distinct structures associated with specific regulators, our results suggest that they are regulated by the same molecular feedback machinery.

# Results

## Two distinct molecular networks appear as coordinated cortical waves

Spontaneous waves have been observed using biosensors such as LimEΔcoil ("LimE", detecting newly polymerized F-actin) and $PH_{crac}$ (detecting PIP3) at the cell cortex of *Dictyostelium*. These waves are better visualized on the basal surface of electro-fused giant cells (Gerhardt *et al*, 2014): $PH_{crac}$ shows as diffuse one-peak bands, while LimE appears as sharp two-peak bands enclosing the edge of $PH_{crac}$ with intermediate intensity in the middle (Fig 1A). Annihilation events when two wave bands meet (Appendix Fig S1A) suggest the excitable nature of the cortical waves. To gain insight into the molecular principles of self-organization, we examined an array of biosensors in waves and found that their morphologies fell into two groups: $PH_{crac}$-like and LimE-like (Fig 1B–E). In the $PH_{crac}$-like group (Fig 1B and C, and Movie EV1), RalGDS (monitoring Rap1 activation) and RBD (monitoring multiple Ras activation) had intensity peaks preceding those of $PH_{crac}$ by $4.11 \pm 1.51$ μm and $1.68 \pm 0.94$ μm, respectively, in the direction of wave propagation, while PKBA (Akt homologue) trailed that of $PH_{crac}$ by $1.28 \pm 0.95$ μm. In the LimE-like group (Fig 1D and E), RacGEF1 (a GEF protein for RacB and Rac1a) showed almost identical patterns as LimE, while coronin had the first peaks in the propagating direction consistently lag those of LimE by $0.32 \pm 0.19$ μm. PAK1-GBD (monitoring Rac1 activation) showed the two-peak morphology, although both intensity peaks at the wave edges were more diffuse than those of LimE. Another Rac activity biosensor, PakB$_{CRIB}$, behaved similarly (Appendix Fig S1B). Different combinations of biosensors in the same cells showed that

these molecular events in waves occurred in a coordinated fashion (Appendix Fig S1C–F). The earliest STEN event we observed, indicated by RalGDS, peaked similarly with CEN sensor LimE but showed a broader profile (Appendix Fig S1F).

In addition to displaying different morphologies in waves, these two groups of biosensors showed distinct dependency on the cytoskeleton. In cells treated with latrunculin A ("LatA", inhibiting actin polymerization), spontaneous $PH_{crac}$ patches were observed, yet PAK1-GBD, from the LimE-like group, appeared completely quiescent, and failed to respond to stimulation of chemoattractant cAMP (Fig 1F and Appendix Fig S2A), as was the case with RacGEF1 (Appendix Fig S2B). In contrast, RalGDS responded to cAMP both with and without the cytoskeleton (Appendix Fig S2C). Taken together, these observations suggest that there are two distinct networks associated with waves (Fig 1G): One includes activated Rap and Ras, PIP3, and PKBs, termed the signal transduction excitable network (STEN); the other is cytoskeleton dependent and includes activated Rac, RacGEF1, F-actin, and coronin, termed the cytoskeleton excitable network (CEN).

## A model of coupled excitable networks simulates the self-organization of the cortical waves

A closer look at the two-peak structure of LimE waves revealed that while the leading band appeared as a continuous structure of F-actin, the trailing band included coarsely spread actin-spots or puncta (Fig 2A and B). These puncta, which lasted $10.8 \pm 3.2$ s, were similar to the undulating actin flashes observed on the cell cortex (Huang *et al*, 2013). None of the STEN biosensors showed puncta-like behavior, suggesting that STEN and CEN have different time scales. To understand how, in principle, the single-peak STEN wave could coexist with the double-banded CEN wave, we turned to a computational model.

To recreate the different dynamics observed for the signaling and cytoskeletal systems, we set up a mathematical model in which STEN and CEN were described by activator–inhibitor systems (Fig 2C, and Materials and Methods). Because CEN waves do not propagate on their own but rather follow the STEN patterns so that their properties are dominated by STEN, STEN was used to control CEN. As previously considered (Huang *et al*, 2013), STEN and CEN have autocatalytic activators ($F_S$ and $F_C$, respectively) and delayed inhibitors ($R_S$ and $R_C$, respectively). To account for the rapid flashes observed on the cell cortex in the absence of STEN activity (Huang *et al*, 2013), the CEN time scale was set to be around eight times faster than STEN's, by decreasing the delay in the CEN inhibitor (high $R_C$; Fig 2C, and Materials and Methods). As a result, simulated CEN activity displayed puncta that did not spread significantly, while STEN waves propagated through the medium (Fig 2D and E).

The interconnection of the two networks ensured that, whereas in the absence of STEN, CEN firings occurred at a low basal level, increasing STEN elevated the basal level of CEN activity. To illustrate the effect of this coupling, three different spatial wave profiles were applied to CEN and the resulting patterns were compared (Materials and Methods). All three profiles had a common leading front but differed in their trailing edge. For an extended stimulus, CEN displayed an elevated level of activity at the leading edge followed by further bands (Appendix Fig S3A). For a spatially restricted stimulus, the initial band was the same but this was not

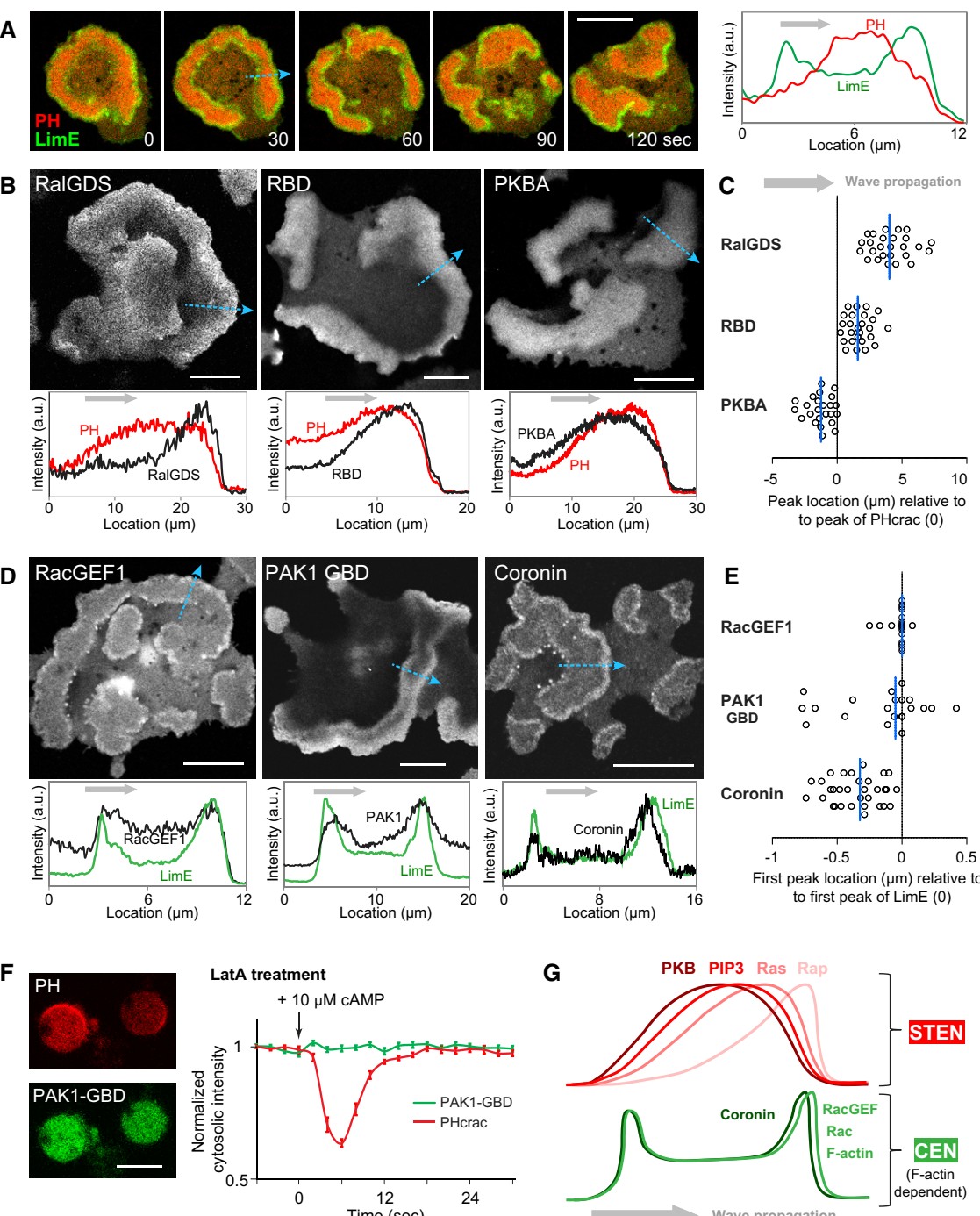

**Figure 1.  Two distinct networks coordinately present as cortical excitable waves.**

A    Left, time lapse confocal images of PH$_{crac}$ (red) and LimE (green) in waves at the basal surface of a giant cell. Right, intensity plot across the blue dotted arrow in image "30 s". Scale bar represents 20 μm.

B–E   Various biosensors in excitable waves. (B, D), snapshots of biosensors (top) and intensity plots (bottom) across the blue dotted arrows, with black plots showing the corresponding biosensors in images above, red (B) showing PH$_{crac}$ in the same cell, and green (D) showing LimE. (C, E), Scatter plots of biosensor peak locations relative to the peak of PH$_{crac}$ (C) or the first peak of LimE (E) across propagating waves (blue solid lines indicate median). Positive values indicate that the peak precedes that of PH$_{crac}$ (or LimE) along the direction of wave propagation. n = 26, 26, 24, 21, 21, 32 cells for RalGDS, RBD, PKBA, RacGEF1, PAK1-GBD, and coronin, respectively. Scale bars in images above represent 20 μm. Gray arrows in all plots represent the direction of wave propagation.

F    Left, confocal images of PH$_{crac}$ (red) and PAK1-GBD (green) in the same cells treated with LatA. Scale bar represents 10 μm. Right, temporal profiles of normalized mean cytosolic intensities of PH$_{crac}$ (red) and PAK1-GBD (green) (mean ± s.e.m., n = 18 cells) in response to cAMP.

G    Cartoon illustrating components and their intensity profiles during wave propagation.

Source data are available online for this figure.

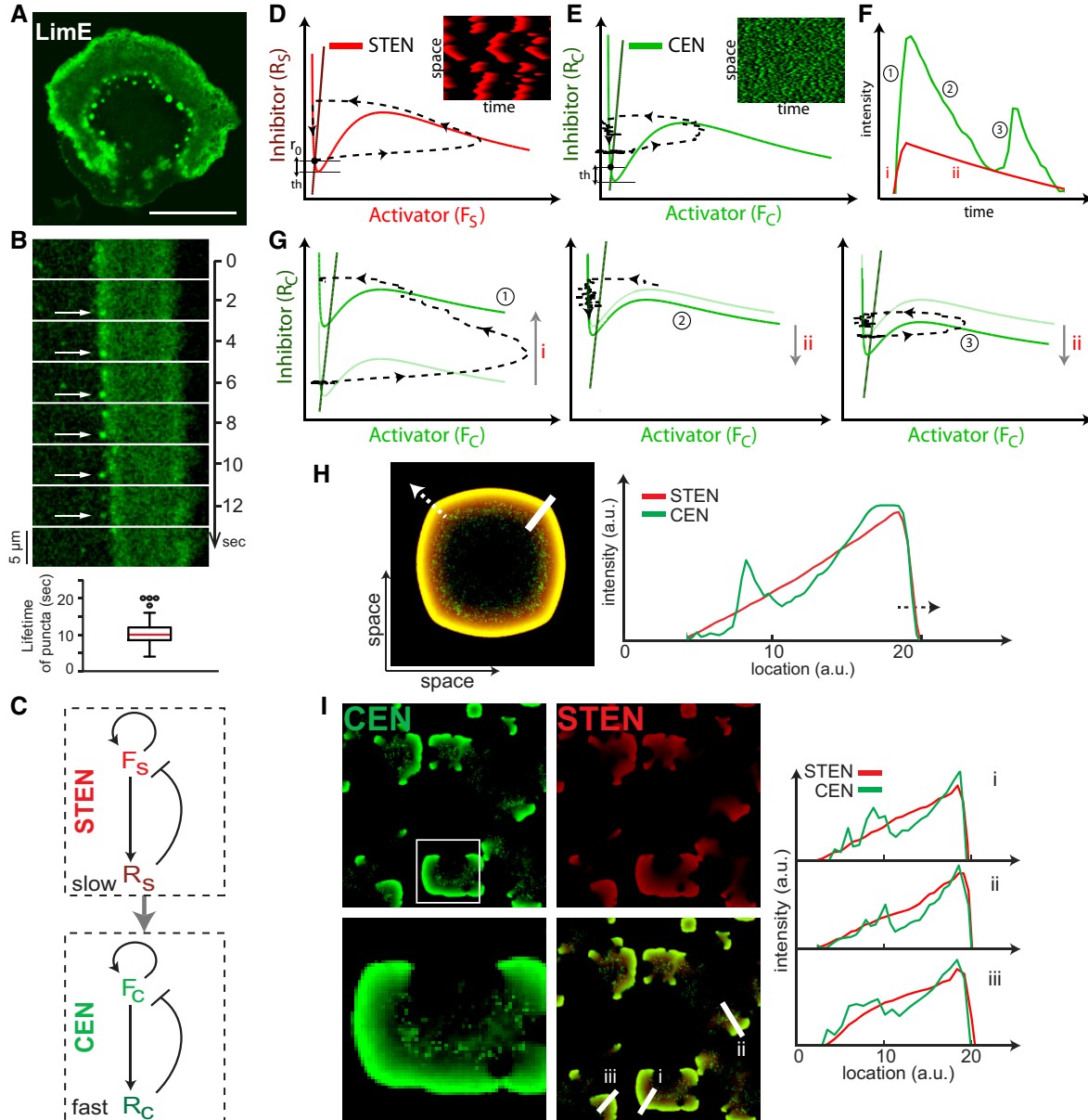

**Figure 2.  A model of coupled excitable networks simulates the self-organization of the cortical waves.**

A   Confocal image of LimE in waves (scale bar, 20 μm).

B   Top, kymograph of LimE in a fixed region across which a piece of wave propagates, with white arrows pointing to the same puncta. Bottom, box plot of puncta's lifetime (*n* = 100 puncta).

C   The coupled STEN-CEN model, with $F_S$ and $R_S$ ($F_C$ and $R_C$) denoting the activator and inhibitor of STEN (CEN), respectively. The interaction between STEN and CEN is shown through the gray arrow.

D   A typical phase plane trajectory of the state in STEN, where the initial inhibition level ($r_0$) and threshold (th) are marked. The inset kymograph shows a typical STEN.

E   A typical trajectory of CEN in phase space showing threshold as in (D). Although the nullclines of STEN and CEN are the same, the trajectory of CEN is significantly shorter resulting in smaller waves as seen from the inset kymograph.

F   A STEN-CEN organization profile with different instances in time marked for the corresponding phase plane trajectories in (G).

G   (left) Plot showing the trajectory of the state in phase plane during the rising profile of STEN (i, from F). The rise of STEN shifts the CEN activator nullcline upwards creating a sharp rise in CEN activity (1, from F). (middle) The initial fall of STEN (ii, from F) resulting in a refractory zone of CEN as its nullcline is lowered (2, from F). (right) As the fall of STEN slows down, CEN reaches a firing position resulting in the second peak of CEN (3, from F).

H   A two-dimensional snapshot of a propagating wave (direction of propagation indicated with the dashed arrow). A line scan corresponding to the STEN-CEN profile is shown on the right.

I   Stochastic simulations of the model in (C), showing the output from STEN (red, top right), CEN (green, top left), and the overlay (bottom right). A zoomed-in version of a CEN wave is shown corresponding to the white box in the CEN. Three line scans are plotted on the bottom right corresponding to the white lines in the overlay panel, showing the double-peak wave architecture of the STEN-CEN model.

Source data are available online for this figure.

followed by any further CEN activity (Appendix Fig S3B). Finally, an intermediate stimulus, chosen to resemble the experimentally observed STEN wave profile, elicited a punctate trailing band following the leading characteristic response (Fig 2F and G). The resulting CEN profile resembled the two-peak structure of LimE observed experimentally (Fig 2H). Although this model result may suggest that STEN should "lead" the CEN waves while experiments indicate otherwise (Fig 1A), we would like to emphasize that the position of the peak is not indicative of the driver of the network. Although STEN controls CEN, it is possible for the STEN peak to be delayed. To illustrate this, we create input profiles by modifying the rise and decay dynamics of STEN. These were fed into CEN so that the input still drives CEN but the peaks are now separated (Appendix Fig S3C and D). For simulation convenience, we use the STEN output profile of Fig 2H for the rest of the paper. We simulated the coupled system in which STEN was triggered stochastically and observed single STEN and double CEN peak patterns that were similar to those seen in experiments (Fig 2I).

## The threshold of excitable networks determines the speed and range of waves

Having established that the STEN and CEN patterns could be recreated, we next considered the kinetics of wave propagation. As expected from an excitable medium, STEN waves annihilate when they collide (Appendix Fig S1A). However, we observed that they can also stop on their own. A typical example of wave stopping is shown in Fig 3A for both signaling (top) and cytoskeletal (bottom) systems as seen in cells. We used theory and simulations to uncover the controlling parameters of wave speed, range, and the mechanism of wave stopping.

The dynamics of the activator ($F$)–inhibitor ($R$) system are best visualized in phase space, where the trajectories of the state reflect the changes in the concentrations of $F$ and $R$ (Fig 2D). A propagating wave is triggered when the system state is displaced sufficiently to cross the threshold for activation. This produces a sharp rise in $F$ without any appreciable increase in $R$, owing to the inherent delay of the inhibitor. This time-scale separation allows wave speed analysis using singular perturbation theory (Keener, 1980; Showalter & Tyson, 1987), which states that the wave speed is determined by the initial level of the inhibitor ($R_0$). Specifically, wave speed is proportional to the net area under the activator nullcline at a particular inhibition level (shaded regions in Fig 3B). A higher system threshold raises $R_0$ and this causes the positive area to decrease (dark red shade, Fig 3B) and the negative area to increase (light red shade, Fig 3B), resulting in a slower wave. At a particular level of inhibition ($R_{stop}$), these two areas are equal, and wave propagation stops.

We used simulations to illustrate a typical example of wave stopping via a kymograph (Fig 3C). After triggering the excitable system at an initial inhibition level, a wave of activity started to propagate. We set the molecular dispersion of the inhibitor to be higher than that of the activator (Materials and Methods). This made the level of inhibition rise in space as the wave propagated, causing the threshold to increase with time in the surrounding medium. This slowed down the wave as it spread in space, reflected by the curving profile seen in the kymograph. By comparing time-profiles (Fig 3C) close to the trigger (i, left) or stopping point (ii, right), we see that in the latter case, the initial inhibition level is closer to $R_{stop}$, indicating

slower wave speed and imminent stopping. In contrast, when the dispersion of the inhibitor was too small, waves propagated indefinitely, as is typical of neuronal waves (Tyson & Keener, 1988; Bhattacharya & Iglesias, 2018a).

Although wave theory states that the level of inhibitor dictates wave velocity, it describes a situation in which the activator parameters are constant. It is actually the relative level of the inhibitor with respect to the minimum of the cubic nullcline (Fig 2D and E) that controls the threshold (Bhattacharya & Iglesias, 2018b). One can change the activator parameters to affect this relative level as well. To illustrate this, we altered the positive feedback strength of the activator—keeping the inhibitor parameters constant—to get similar wave stopping and wave speed results (Appendix Fig S3E and F). As reflected in the parameter table, the positive feedback strength is more sensitive as compared to the negative feedback.

Based on these simulations, we made two predictions of how wave characteristics change when the threshold is altered. First, at a lower threshold, the wave will be faster because of the lower initial level of $R$ (Fig 3D). Second, the wave will propagate further because the initial inhibitor level will be further away from the critical wave stopping point, $R_{stop}$ (Fig 3D, right). We illustrated these principles through two-dimensional stochastic simulations in which waves were triggered with high and low thresholds and the subsequent propagations were compared (Fig 3E). Although the two waves started together (yellow spot in first frame), by the last frame, the wave generated by the system with higher threshold had propagated more slowly and had also broken up owing to reduced wave range. This inverse correlation between threshold and wave speed is observed in a particular range of threshold (Appendix Fig S3G), beyond which the system either has no appreciable wave spread (high threshold) or oscillates synchronously (low threshold). Finally, we showed that these principles depend on a combination of diffusion and time delay of the inhibitor and not solely on diffusion coefficient (Appendix Fig S3H).

## Perturbations to STEN reveal its molecular architecture

To uncover the molecular basis of STEN, we next considered altering the system both through simulations, by changing the strength of various feedback loops, and experiments, by introducing specific perturbations. Though our model using one-way coupling captures the organization of the STEN-CEN waves (Fig 2C), other observations suggest that CEN also feeds back to influence STEN triggering (Huang *et al*, 2013; Taniguchi *et al*, 2013). Previously, we modeled the feedback from CEN to STEN as having fast, positive and slow, negative components (Shi *et al*, 2013; Materials and Methods). As shown in Fig 4A, we introduced these elements into the STEN-CEN model used here.

We lowered STEN threshold by increasing the strength of the positive feedback, simulated the system, and observed the effect on CEN waves (Appendix Fig S3I). As expected, STEN waves spread further, causing the coupled CEN to follow suit (Fig 4B and Movie EV2). A three-dimensional visualization (Fig 4C) showed that, initially, the waves had smaller range and broke up easily. After lowering the threshold, the waves propagated further in space with a significantly faster speed (Fig 4D).

We tested this in cells by using the chemically inducible dimerization (CID) system for acute perturbations. Recruiting Inp54p by

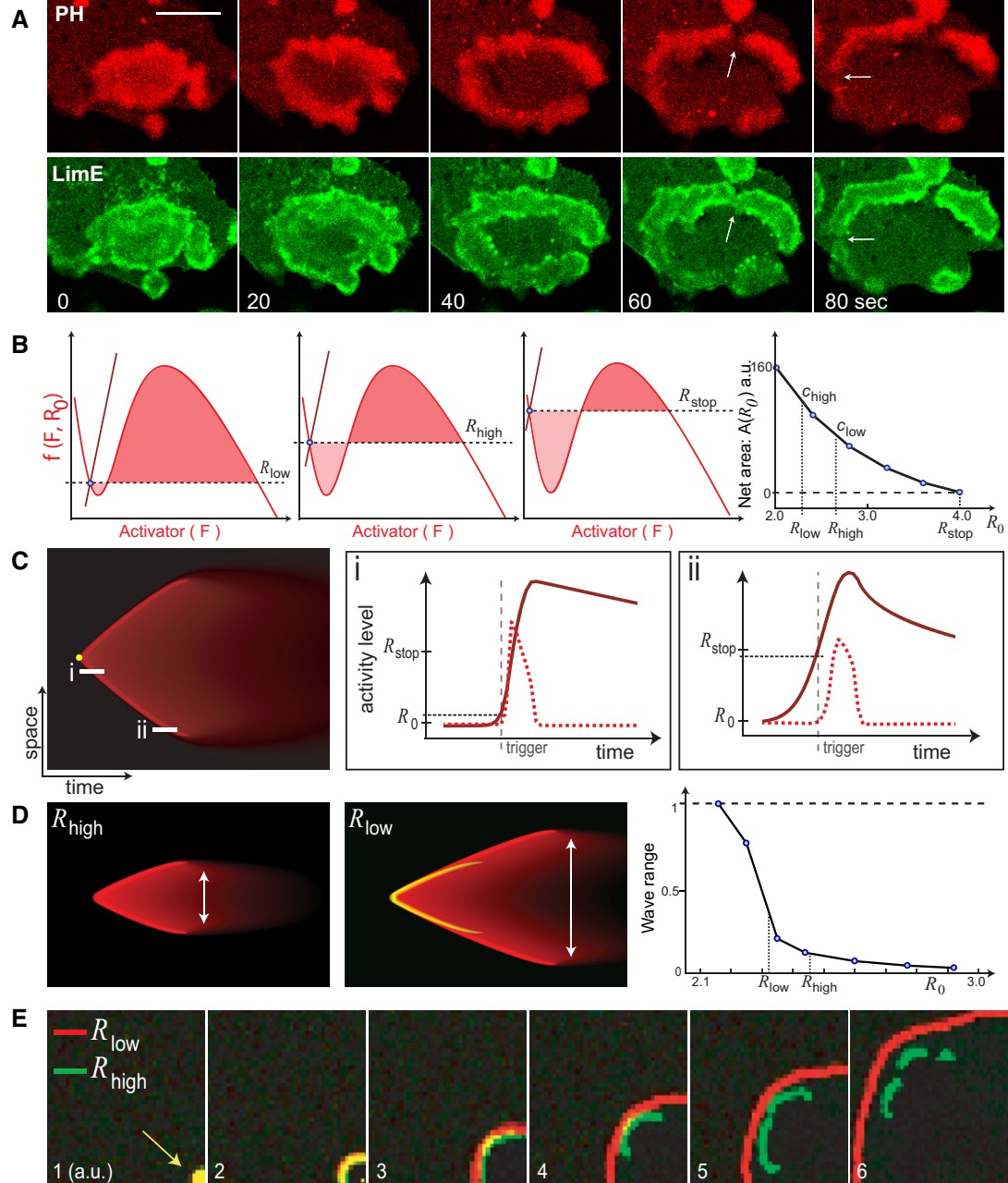

**Figure 3.  The threshold of excitable networks determines the speed and range of waves.**

A   Time lapse confocal images of $PH_{crac}$ (top) and LimE (bottom) in the same giant cell (scale bar, 20 μm). White arrows point to wave breaking events.

B   Three phase plane representations of the activator (*F*)–inhibitor (*R*) system with the activator nullcline (red) and the inhibitor nullcline (brown) for different inhibitor levels. The horizontal dashed line corresponds to the initial inhibition level ($R_0$) at equilibrium (blue circle). The net area (rightmost plot) is calculated by adding the positive area above the horizontal line (dark red shade) and the negative area below (light red shade), for increasing levels of inhibition ($R_0 = R_{low}$, $R_{high}$, and $R_{stop}$). The points $R_{low}$ and $R_{high}$ correspond to high ($c_{high}$) and low ($c_{low}$) wave velocities, respectively, as indicated on the right.

C   (Left) Kymograph plotting the inhibitor activity with time and space as axes. The point where the wave is triggered is indicated in yellow. (Right) Two line scans (i and ii) from the kymograph and the corresponding activity levels. $R_0$ indicates the inhibition level at the time of the first trigger, while $R_{stop}$ indicates the critical stopping threshold. The activator profile is shown via the dashed red plot, while the inhibitor is the solid brown line. The gray vertical dashed line marks the point when the activator element is triggered. The black horizontal dashed line denotes the inhibition level at the time of the specific trigger shown.

D   Kymographs corresponding to high (left) and low (middle) thresholds with the white arrow denoting the range of the wave, which is calibrated with threshold $R_0$ (right). The high threshold wave is superimposed in yellow on the low threshold wave to emphasize the increase in wave velocity as the low threshold wave spreads further in space compared to the high threshold wave.

E   Two-dimensional simulations where two waves, initiated at different thresholds, are triggered at the bottom left corner (leftmost panel). Overlap between the two waves appears as yellow. The panels show the evolution of the two waves with time.

Source data are available online for this figure.

the CID system to reduce PIP2 levels was shown to successfully lower the threshold for STEN activation (Miao *et al*, 2017), and we further examined its effects on waves. Within minutes following rapamycin-induced Inp54p recruitment, nascent patches of LimE waves started to expand outwardly in all directions until reaching the edge of the cell or annihilating upon meeting adjacent waves (Fig 4E and Movie EV3). Compared with waves beforehand, which had limited range and asymmetric expansion patterns, waves after PIP2 reduction typically propagated symmetrically without stopping (Fig 4F and G). Interestingly, the second LimE peak became more diffuse (Fig 4G). After about 10 min following rapamycin, the triggering of waves synchronized and the waves merged, turning the cell into a giant "oscillator" (Appendix Fig S4A). Similar effects were observed using STEN biosensor PH$_{crac}$ (Appendix Fig S4B and Movie EV4). We compared wave speed before and after PIP2 reduction using an automatic algorithm and found a significant increase compared with controls (Fig 4H and I). Further, the expansion of STEN waves (PH$_{crac}$) after PIP2 reduction even occurred under LatA treatment (Fig 4J and Movie EV5), suggesting that PIP2 levels mediate a direct role in STEN independent of the cytoskeleton.

Elevating Rap or Ras activities through various nodes by the CID system also led to modest increases in wave speed (Appendix Fig S4C–F). In these cases, the second peak of LimE remained sharp (Appendix Fig S4G). These suggest that Rap/Ras and PIP2 have antagonizing roles in STEN activities. Furthermore, while Rap/Ras biosensors labeled active wave zones (Fig 1B), previous reports showed that PIP2 biosensor PH$_{PLC\delta}$ fell off from those regions (Gerisch *et al*, 2011). Consistently, we found that PI5K, which synthesizes PIP2, showed a complementary pattern to STEN/CEN waves (Fig 4K). Taken together, our results suggest that PIP2 levels and Rap/Ras activities inhibit each other to constitute a positive feedback loop that controls the threshold for STEN activation (Fig 4L).

## PKBs provide negative feedback in STEN and couple STEN to CEN

To gain insight into the delayed negative feedbacks in STEN, we turned to PKBs, which are suggested to negatively regulate Ras activities (Charest *et al*, 2010) and showed a delayed profile during wave propagation (Fig 1B and C). Acute recruitment of cytosolic PKBA to plasma membrane was sufficient to activate it and its substrates while also increasing the activity of endogenous PKBs (Appendix Fig S5A and B). These activities were inhibited by the combination of LY294002 (PI3K inhibitor) and PP242 (TorC2 inhibitor).

Dramatically, increasing PKB activities rapidly caused a significant decrease in the speed and range of the LimE waves, turning them into tiny patches (Fig 5A–C and Movie EV6). At the same time, the number of patches increased significantly (Fig 5C), consistent with the conventional role of PKBs in mediating cytoskeletal events. Similarly, other STEN and CEN biosensors, including PH$_{crac}$, RalGDS, and RacGEF1, rearranged into non-propagating patches (Fig 5D and Appendix Fig S6). Importantly, the relative relationship between STEN and CEN biosensors was preserved in these patches, where PH$_{crac}$ had highest intensity in the center while LimE at the edge (Fig 5E and F, and Movie EV7). To examine the role of PKBs in STEN directly, we used LatA to exclude contributions of F-actin in this context. Recruiting PKBA quickly quenched spontaneous STEN activities represented by the biosensor PH$_{crac}$ (Fig 5G, Appendix Fig S5C, and Movie EV8). In addition, the same approach dampened the response of PH$_{crac}$ to an unsaturated dose of cAMP stimulation under LatA treatment (Appendix Fig S5D). These together suggest that recruiting PKBA led to an elevated threshold for STEN activation.

These observations indicate that increasing PKB activity has two effects (Fig 5G). On one hand, it mediates negative feedback, as indicated by the decrease of wave speed and range, thus raising the threshold for STEN activation. On the other hand, it increases the basal level of input to CEN, thus lowering the threshold for CEN firings, as illustrated by the greater number of patches. We tested this scheme (Fig 5H) through simulation by raising the STEN threshold and lowering the CEN threshold simultaneously (Appendix Fig S3J). This resulted in the appearance of a large number of waves with small range, which closely resembled those observed in experiments (Fig 5I and Movie EV9). The CEN activity was still organized by the STEN waves, which had broken up into smaller wavelets (Fig 5J). These waves were significantly smaller and slower, consistent with high threshold predictions (Fig 5K and L). While raising the STEN threshold would typically lower the number of firings, this was

**Figure 4. Perturbations to STEN reveal its molecular architecture.**

A   The STEN-CEN model architecture where $F_s$ and $R_s$ ($F_c$ and $R_c$) denote the activator and inhibitor of STEN (CEN), respectively, showing the feedback connections between CEN and STEN. The positive feedback is indicated through the arrowhead, while the bar at the end of the arrow refers to negative feedback.

B   Time lapse images showing simulations of the coupled STEN-CEN model demonstrating CEN (top) and STEN (bottom) waves spreading in space. The threshold was lowered by increasing positive feedback of STEN (perturbation) at time 0. The white arrow (time 0, CEN) denotes a wave at a high threshold that disappears in the next image, while the red arrow (time 25, CEN) points to a lowered threshold wave that enlarges and spreads.

C   A 3D representation of simulation waves under high (left) and low (right) threshold waves propagating in time.

D   Quantification of simulated wave speed before and after threshold lowering (+perturb). Statistics are from eight simulations. Error bars are from the Student's *t*-test.

E   Time lapse confocal images of LimE at the basal surface of a giant cell, which recruits Inp54p to membrane induced by rapamycin at time 0. The white arrow (−45 s) points to a nascent wave before Inp54p recruitment, while the red arrow (60 s) after Inp54p recruitment.

F   Color-coded overlays of LimE before and after Inp54p recruitment. White arrows point to nascent waves at time 0.

G   Kymographs of LimE in regions outlined by dotted line boxes in (F).

H, I   Box plots of mean wave speed (left) and fraction of fastest pixels (right) before and after Inp54p (H) or FRB (I) recruitment (red bars indicate median, *n* = 25 cells each). Error bars are from the Student's *t*-test.

J   Time lapse confocal images of PH$_{crac}$ at the bottom of giant cells under LatA treatment, where rapamycin is added at time 0 to induce Inp54p recruitment. White arrows point to wave patches right before rapamycin addition.

K   Confocal image of PI5K (left) and LimE (right) at the bottom of the same cell.

L   Cartoon highlighting positive feedbacks in STEN.

Data information: All scale bars in images represent 20 μm. (D, H) Boxes indicate upper and lower quartile; whiskers indicate maximum and minimum.
Source data are available online for this figure.

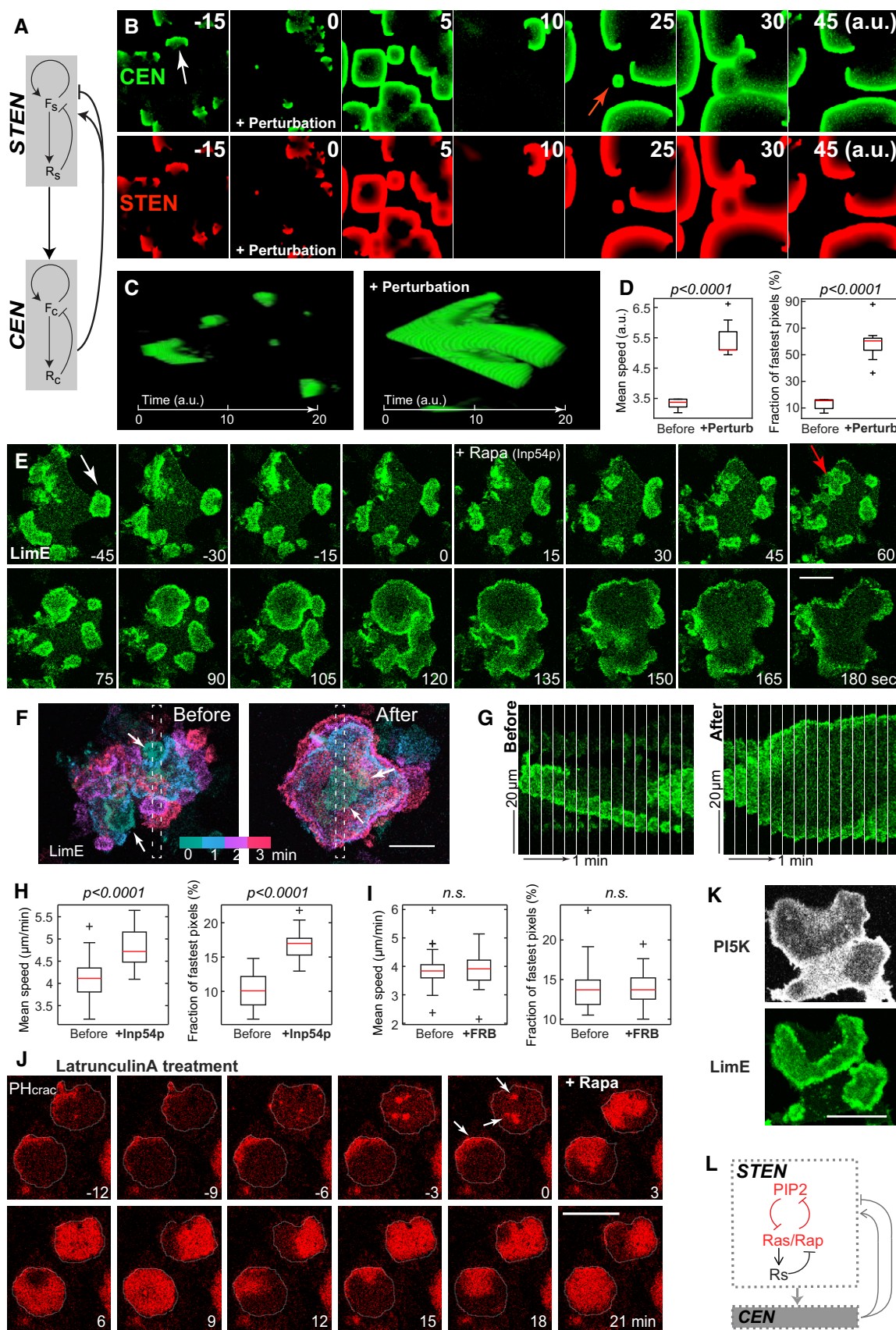

**Figure 4.**

compensated by the increase in CEN activity, which feeds back positively onto STEN. Though CEN also provides negative feedback to STEN in our model (Fig 4A), the slower dynamics of this feedback filter out the fast CEN activities (Appendix Fig S3I).

### Perturbations uncover the molecular architecture of CEN and its feedback to STEN

To illuminate the organizing principles of waves further, we developed tools to change the threshold of CEN directly. Using the CID system, we acutely recruited the catalytic domain of RacGEF1 (Park *et al*, 2004), RacGEF1$^{\Delta N}$, to the plasma membrane, and found that it triggered massive actin polymerization events at the cell periphery within minutes (Fig 6A and Movie EV10). In giant cells, the typically organized LimE wave patterns were totally altered. In addition to LimE signals detected along the whole perimeter, random diffuse patches and puncta took over the cell bottom (Fig 6B–D and Movie EV11). Interestingly, these LimE puncta have a lifetime of $10.5 \pm 3.5$ s (Fig 6D), similar to the puncta observed at the second peak of LimE during wave propagation (Fig 2B). Given that full-length RacGEF1 co-localized with LimE in waves (Fig 1D), a positive feedback loop consisting of Rac activities and F-actin was clear (Fig 6E).

These observations suggest that recruiting RacGEF1$^{\Delta N}$ lowered the threshold of CEN, and we next tested this effect in our model. Since the resulting observed diffused patches were longer-lived than the aforementioned puncta (Fig 6D), suggesting a state with greater activity, we increased the positive feedback of CEN to account for the perturbation (Fig 6E) shifting the equilibrium to a higher stable state (Appendix Fig S3K). Simulations following this adjustment showed large patches of CEN (Fig 6F, top; Movie EV12), consistent with those seen experimentally. These patches were diffuse and lacked organization (Fig 6G), suggesting that STEN was being inhibited. Simulations captured an initial burst due to the fast, positive feedback from CEN to STEN, followed by a total abrogation of STEN activity (Fig 6F, bottom; Movie EV12), as the slow, negative feedback from CEN ensued. The global triggering of the negative feedback in this case led to a stronger effect accounting for the total shut-off of STEN, in contrast to the more local effect seen previously (Fig 5I). This negative feedback remained elevated and rendered the system impervious to the continuously changing patch dynamics (Appendix Fig S3K).

When investigating STEN activities in cells, we found that PH$_{crac}$ signals were inhibited after RacGEF1$^{\Delta N}$ recruitment (Fig 7A and Movie EV13). In a cell with both PH$_{crac}$ and LimE as readout, the difference was obvious: While LimE greatly increased at the cell perimeter and displayed as patches and puncta, PH$_{crac}$ was completely quiescent (Fig 7B,C). The different responses of STEN and CEN following RacGEF1$^{\Delta N}$ recruitment were further demonstrated by biosensors RalGDS (Fig 7D and Appendix Fig S7A and B) and full-length RacGEF1 (Appendix Fig S7C and D). Here, in the absence of STEN activities, CEN components abandoned their signature morphology (Fig 1G) but rather displayed the highest intensity in the center of patches. These results further support STEN's organizing role of CEN.

Careful examination showed that STEN activities were promoted transiently before being inhibited, like those seen in our simulations. At the cortex of single cells, PH$_{crac}$ intensity went up in the first 2 min during RacGEF1$^{\Delta N}$ recruitment before staying at a reduced level after 5 min (Fig 7E–G). Effects of RacGEF1$^{\Delta N}$ recruitment were lost in LatA treatment (Fig 7H), suggesting that these responses of STEN relied on the cytoskeleton. Moreover, the steady inhibiting effects on STEN could be overcome by an unsaturated dose of chemoattractant (Fig 7I). These results support that CEN has both a fast, promoting and a slow, inhibiting effect on STEN, validating our computational model. To look for molecular mediators of these feedbacks, we examined F-actin-binding GflB and RapGAP1, which are GEF and GAP proteins for Rap1, respectively. These proteins showed LimE-like patterns in waves, with RapGAP1 lagging GflB and LimE (Fig 7J–L). Further, GflB and RapGAP1 displayed diffuse patches and signals along cell periphery after RacGEF1$^{\Delta N}$ recruitment (Fig 7M and N), similar with previous CEN biosensors. These observations suggest that GflB and RapGAP1, which are under CEN regulation but mediate STEN activities, serve as positive and negative feedbacks, respectively, from CEN to STEN (Fig 7O).

### Cortical wave patterns correlate with profiles of cellular protrusion

Cortical waves have been suggested to play roles in cell migration. We showed previously that lowering the threshold of STEN promoted broad, sheet-like expansions over small, cup-like

---

**Figure 5.  PKBs provide negative feedback in STEN and couple STEN to CEN.**

A   Time lapse confocal images of LimE at the basal surface of a giant cell, which recruits PKBA to membrane induced by rapamycin at time 0.
B   T-stack of LimE in the region outlined by the colored line box in (A).
C   Box plots of mean wave speed (left), mean wave patch area (middle), and average number of wave patches per frame (right) before and after PKBA recruitment (red bars indicate median, *n* = 24 cells). Error bars are from the Student's *t*-test.
D   Time lapse confocal images of PH$_{crac}$ at the basal surface of a giant cell, which recruits PKBA to membrane induced by rapamycin at time 0.
E   Confocal images of PH$_{crac}$ (left) and PH$_{crac}$ merged with LimE (middle, PH$_{crac}$ as red, LimE as green) in a giant cell after steady PKBA recruitment. Right, intensity plots of PH$_{crac}$ (red) and LimE (green) across the blue box.
F   Kymograph of PH$_{crac}$ (red) merged with LimE (green) along the white dashed line from (E).
G   Time lapse confocal images of PH$_{crac}$ at the bottom of a giant cell treated with LatA, where rapamycin is added at time 0 to induce PKBA recruitment.
H   Cartoon highlighting PKB mediating negative feedbacks in STEN and giving inputs to CEN.
I   Time lapse images showing simulations (CEN—top, STEN—bottom) of the model in g with the feedbacks in red altered at time 0 (perturbation).
J   (top) An example of an overlaid STEN-CEN wave with STEN in red and CEN in green. The line scan (blue box) is shown below.
K   A 3D representation of the CEN waves before and after the perturbation is introduced (time 0).
L   Quantification of different wave characteristics before and after the perturbation. Error bars are with eight simulations, from the Student's *t*-test.

Data information: All scale bars represent 20 μm. (C, L) Boxes indicate upper and lower quartile; whiskers indicate maximum and minimum.
Source data are available online for this figure.

---

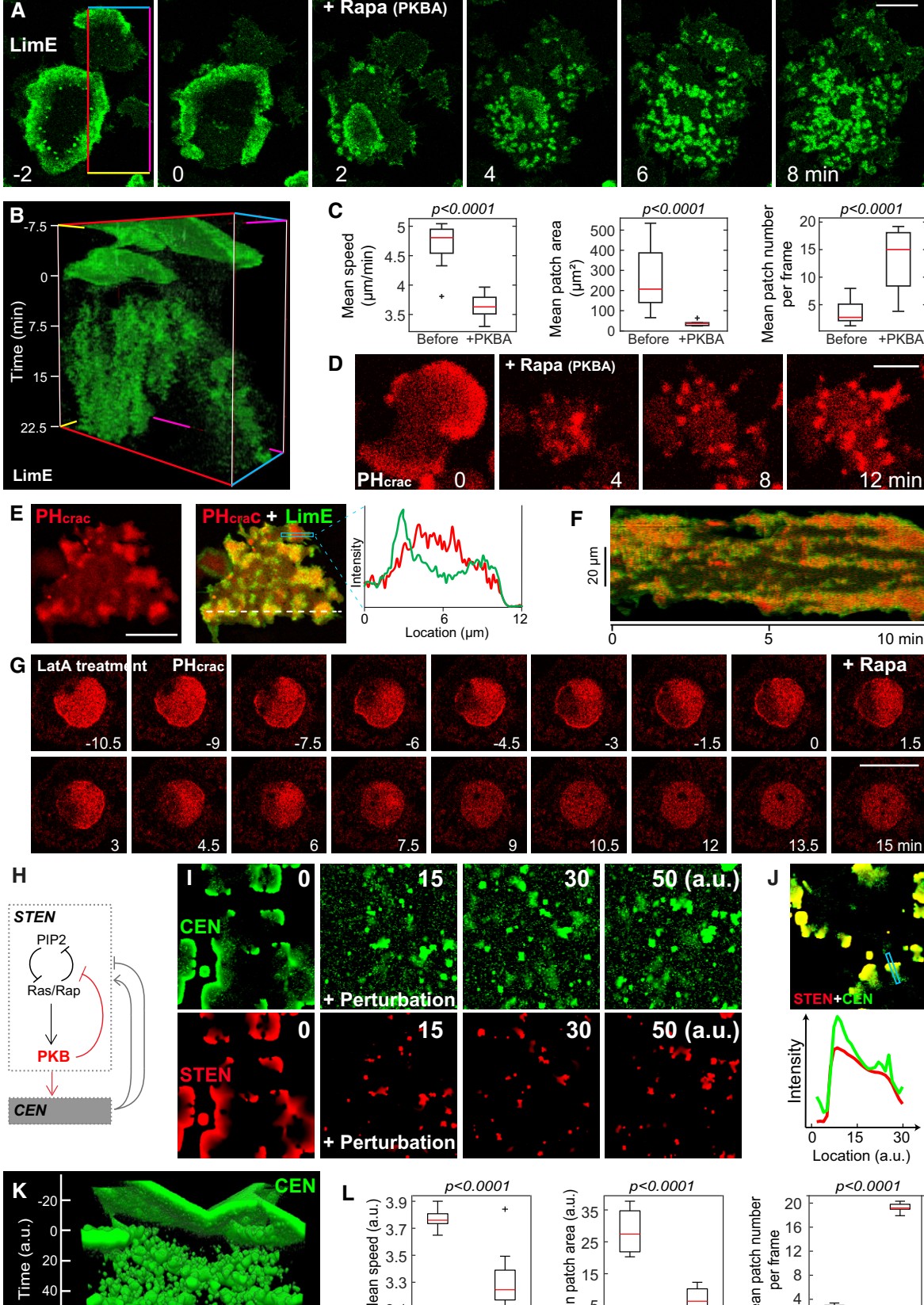

**Figure 5.**

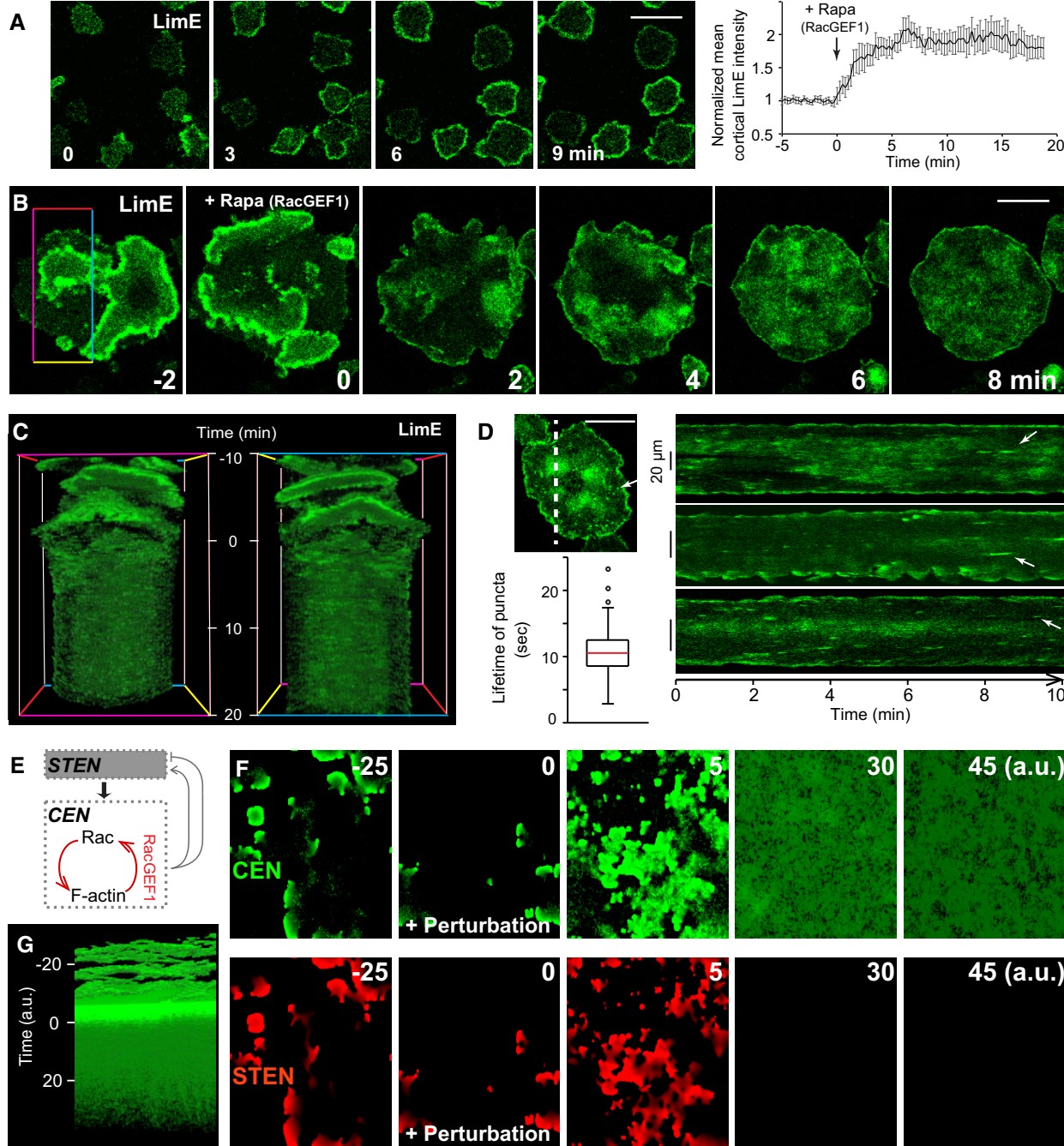

**Figure 6.  RacGEF1-mediated positive feedbacks in CEN modulate wave profiles.**

A   Left, time lapse confocal images of LimE in single cells following RacGEF1[ΔN] recruitment, induced by rapamycin addition at time 0. Right, temporal profile of normalized mean LimE intensity at cell cortex (mean ± s.e.m., *n* = 20 cells).

B   Time lapse confocal images of LimE at the basal surface of a giant cell, which recruits RacGEF1[ΔN] to membrane induced by rapamycin at time 0.

C   T-stack of LimE in the region outlined by the colored line box in (B).

D   Right, kymographs of LimE along a line across giant cells after RacGEF1[ΔN] recruitment. Top left, confocal image with a dotted line as an example used to make kymographs. White arrows in image and kymographs point to examples of actin puncta. Bottom left, box plot of lifetime of puncta (red bar indicates median, *n* = 100 puncta from 10 cells). Boxes indicate upper and lower quartile; whiskers indicate maximum and minimum.

E   Cartoon highlighting positive feedbacks in CEN.

F   Simulations of the model in (E), showing the CEN (top) and STEN (bottom) activity, before and after the feedback is altered (perturbation) at time 0.

G   The CEN activity stacked in time showing how the waves become diffuse after the perturbation is introduced.

Data information: All scale bars represent 20 μm.
Source data are available online for this figure.

protrusions, causing amoeboid cells to transition to keratocyte-like and oscillatory cells (Miao *et al*, 2017). Acutely recruiting PKBA, which elevated the threshold of STEN and concomitantly increased the input into CEN, generated cells with long, thin protrusions all over the perimeter and hindered motility (Fig 8A and Movie EV14). On the other hand, recruiting RacGEF1[ΔN], which drastically lowered the threshold of CEN and caused STEN inhibition, rendered cells immobile with ruffle-like protrusions along the whole perimeter (Fig 8B and Movie EV14). Although cells after RacGEF1[ΔN] recruitment had persistently high level of actin polymerization, they failed to form any big protrusions and did not spread as dramatically as oscillators (Fig 8C), supporting the notion that CEN alone is not sufficient to form sustained protrusions for cell motility. Images of phalloidin-stained cells illustrating the diverse protrusions following perturbations are shown in Fig 8D. Simulations of our computational model coupled to a viscoelastic mechanical model, under various perturbations, captured the experimentally observed cellular protrusions (Fig 8E and Movie EV15).

## Discussion

All together, we combined theoretical and experimental approaches to reveal a molecular architecture controlling the organization of cortical waves (Fig 9A). We found that Ras/Rap- and Rac/actin-centered networks, STEN and CEN, respectively, interact to control dynamic wave patterns at the cell cortex. Although the components we highlight are only a subset of the overall system (Devreotes *et al*, 2017; van Haastert *et al*, 2017; Fort *et al*, 2018; Li *et al*, 2018; Tanabe *et al*, 2018), they capture essential features of wave organization. In each network, various components sequentially and coordinately engage in wave propagation. Perturbations on each of these constituents have dramatic effects, suggesting that they are all integral parts of the excitable networks. In each of our perturbations, a specific component of the network was "clamped". Rather than causing the

whole network to become similarly clamped, as would occur in a simple cascade connection, the intricate feedback connections within the coupled networks allowed the system to adjust by organizing other elements into new dynamic patterns. The new patterns caused by each of these perturbations correspondingly led to distinct cellular protrusions involved in migration (Fig 9B), leading to a network-centric theory on the generation of diverse cortical protrusions.

### Small GTPases centric networks closely interact to bring about wave patterns

In our molecular scheme (Fig 9A), we propose that PIP2 and Ras/Rap activities constitute a positive feedback loop in STEN, in which lowered PIP2 levels promote more Ras/Rap activities possibly via regulating GEF and GAP proteins, and elevated Ras/Rap can further lower PIP2 levels through the activation of PI3K and PLC. Once the positive feedback loop starts, PKBs' activation can be set off in a delayed manner and serve as a negative feedback loop by elevating PI5K activity to increase PIP2 synthesis (Kamimura *et al*, 2008; Fets *et al*, 2014) and inhibiting Sca1-associated GEF-containing complex (Charest *et al*, 2010). Through another set of substrates, PKBs transmit information to CEN, where Rac activity and F-actin comprise a positive feedback loop through F-actin binding RacGEF1, and where time-delayed coronin could engage in a negative feedback loop. CEN also regulates STEN through F-actin-dependent GflB and RapGAP1, which are GEF and GAP proteins, respectively, for Rap GTPase. Together, STEN and CEN are coordinately linked with each other, such that supra-threshold fluctuations or inputs in any component can trigger the whole system leading to wave propagation.

Although linked, STEN and CEN play fundamentally different roles in wave organization. Components of each network displayed distinct morphologies and dynamics, with STEN showing diffuse bands and CEN displaying sharp bands with puncta. Our simulations showed that a difference in time scales, STEN as a slow, and CEN as a fast, excitable network, is sufficient to recreate the

**Figure 7. CEN exerts fast positive and slow negative feedbacks to STEN.**

A   Time lapse confocal images of PH[crac] at the basal surface of a giant cell, which recruits RacGEF1[ΔN] to membrane at time 0 induced by rapamycin.
B   Confocal images of PH[crac] (left) and PH[crac] merged with LimE (right, PH[crac] as red, LimE as green) in a giant cell after RacGEF1[ΔN] recruitment.
C   Kymograph of LimE (top, green) and PH[crac] (bottom, red) across the dotted line in (B).
D   Confocal image of RalGDS (left) in a giant cell after RacGEF1[ΔN] recruitment and the corresponding kymograph (right) across the dotted line in the image.
E   Time lapse confocal images of PH[crac] in single cells, which recruit RacGEF1[ΔN] to membrane induced by rapamycin at time 0. White arrow points to PH[crac] patches at cell cortex.
F   Kymographs of PH[crac] at cell cortex before and after RacGEF1[ΔN] recruitment.
G   Temporal profiles of normalized PH[crac] intensities at cell cortex before and after RacGEF1[ΔN] recruitment (mean ± s.e.m., *n* = 20 cells). Red dashed lines in (F, G) indicate rapamycin addition at time 0.
H   Distribution of different sizes of PH[crac] patches at cell cortex under LatA treatment, with (solid) or without (hollow) RacGEF1[ΔN] recruitment (~ 6,000 cells from three independent experiments).
I   Temporal profiles of normalized cytosolic intensities of PH[crac] in response to 1 nM cAMP stimulation at time 0, with (red) or without (blue) RacGEF1[ΔN] recruitment (mean ± s.e.m., *n* = 25 cells each).
J   Confocal images of GflB (top) and RapGAP1 (bottom) in waves at the basal surface of a giant cell.
K   Intensity plots of GflB (blue) and RapGAP1 (orange) across the green dotted arrow in (J).
L   Scatter plots of first peak distances, of GflB relative to LimE, RapGAP1 relative to LimE, and RapGAP1 relative to GflB in the same giant cells (red lines indicate median, *n* = 19 cells each).
M   Confocal images of GflB (top) and RapGAP1 (bottom) at the basal surface of the same giant cell after RacGEF1[ΔN] recruitment.
N   Kymographs of GflB (top) and RapGAP1 (bottom) across the green dotted lines in (M).
O   Cartoon highlighting positive and negative feedback loops from CEN to STEN.

Data information: All scale bars represent 20 μm.
Source data are available online for this figure.

    

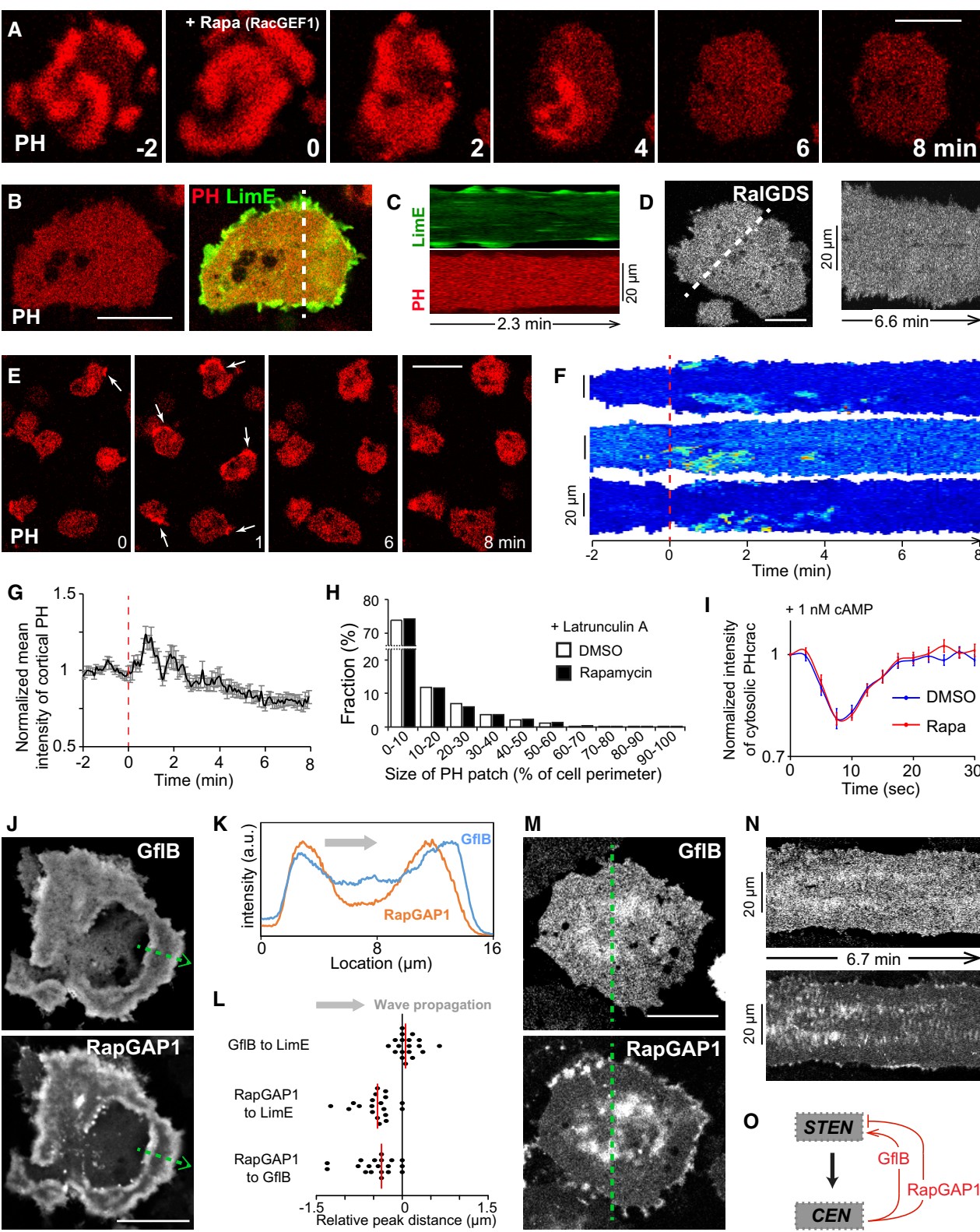

**Figure 7.**

observed morphologies. Perturbations on STEN components, including PIP2, Ras, and PKBA, drastically changed wave characteristics. Notably, in all these cases, CEN components retained their signature morphologies and relative relationship with STEN markers. Only in the case where STEN was totally inhibited due to strong negative feedback after recruiting RacGEF1^ΔN, CEN components lost their

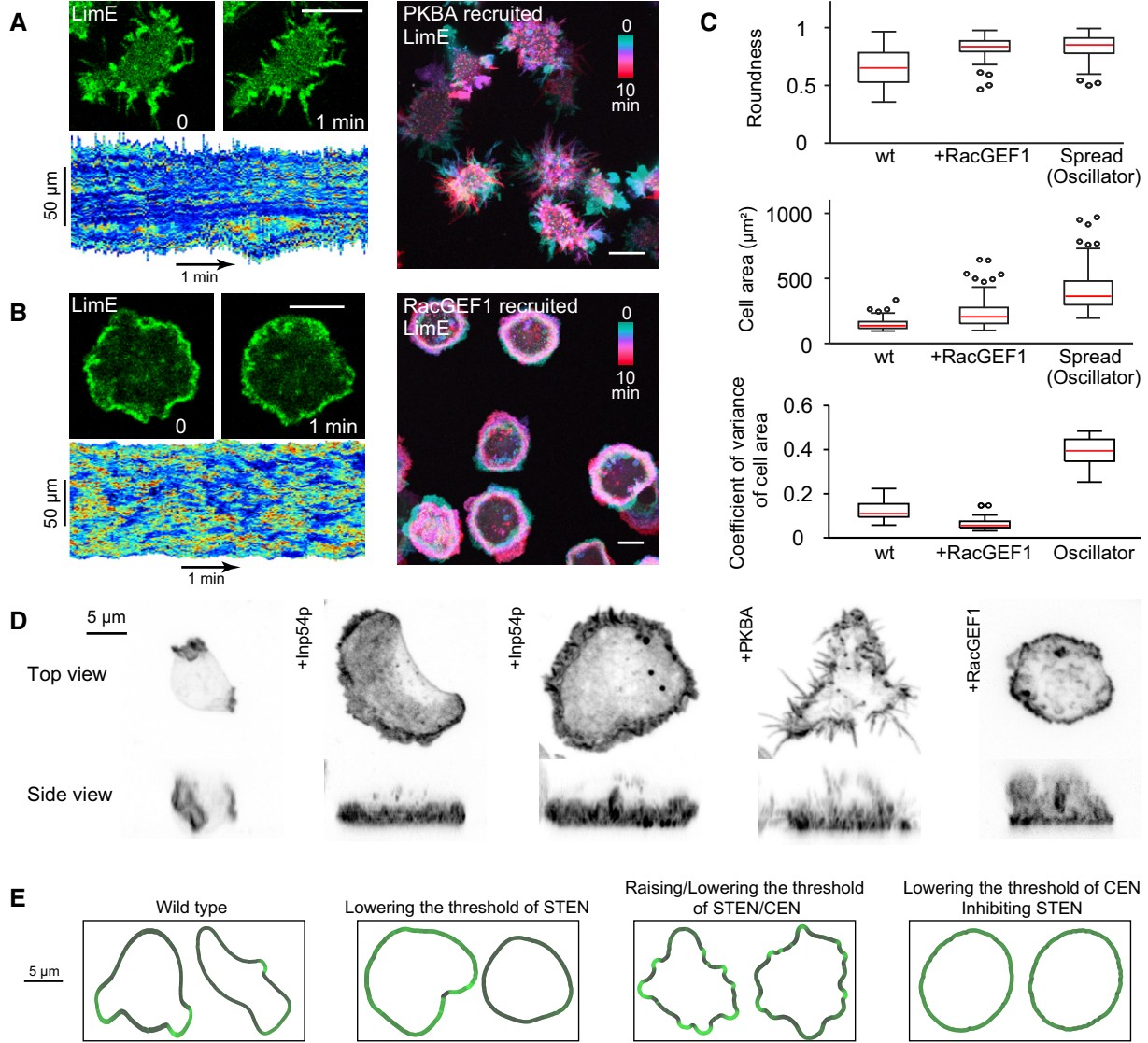

**Figure 8.  Perturbations causing different cortical patterns also altered cellular protrusions correspondingly.**

A, B    Left, confocal images of LimE (top) and kymographs of cortical LimE (bottom) in single cells with PKBA (A) or RacGEF1$^{\Delta N}$ (B) steadily recruited (scale bars, 10 μm). Right, color-coded overlays of LimE in single cells.

C    Box plots of roundness (left) and cell area (middle) comparing wt, cells with RacGEF1$^{\Delta N}$ recruited, and spreading phases of oscillators after Inp54p recruitment ($n = 100$ cells each). Right, box plots of coefficient of variance of cell area with the above conditions in a 10-min time window ($n = 30$ cells each). Red bars indicate median. Boxes indicate upper and lower quartile; whiskers indicate maximum and minimum.

D    Phalloidin staining of cells after perturbations. Maximal intensity projections are shown.

E    Snapshots from level set simulations using the activity obtained from earlier simulations.

Source data are available online for this figure.

organized morphology and displayed as diffuse patches and random puncta. Taken together, these observations suggest that STEN orchestrates CEN and dictates the wave speed and range, while CEN controls the triggering of STEN through feedbacks.

### Feedback strengths of the network control speed and range of waves

There are multiple parameters determining wave characteristics. First, the cortical waves observed appear as traveling pulses with trailing refractory zones, rather than only advancing fronts. The difference between these two types of waves is that fronts can arise from single-species dynamics, but pulses require at least two interacting components (Holmes *et al*, 2012). This suggests that these waves arise from an activator–inhibitor structure operating in the excitable regime, indicating the need for at least two interacting components as well as diffusion. Second, although diffusion is necessary for a wave to propagate, it does not appear to be rate limiting since the reaction rates are much slower than diffusion of ions/lipids/proteins between neighboring cortical elements.

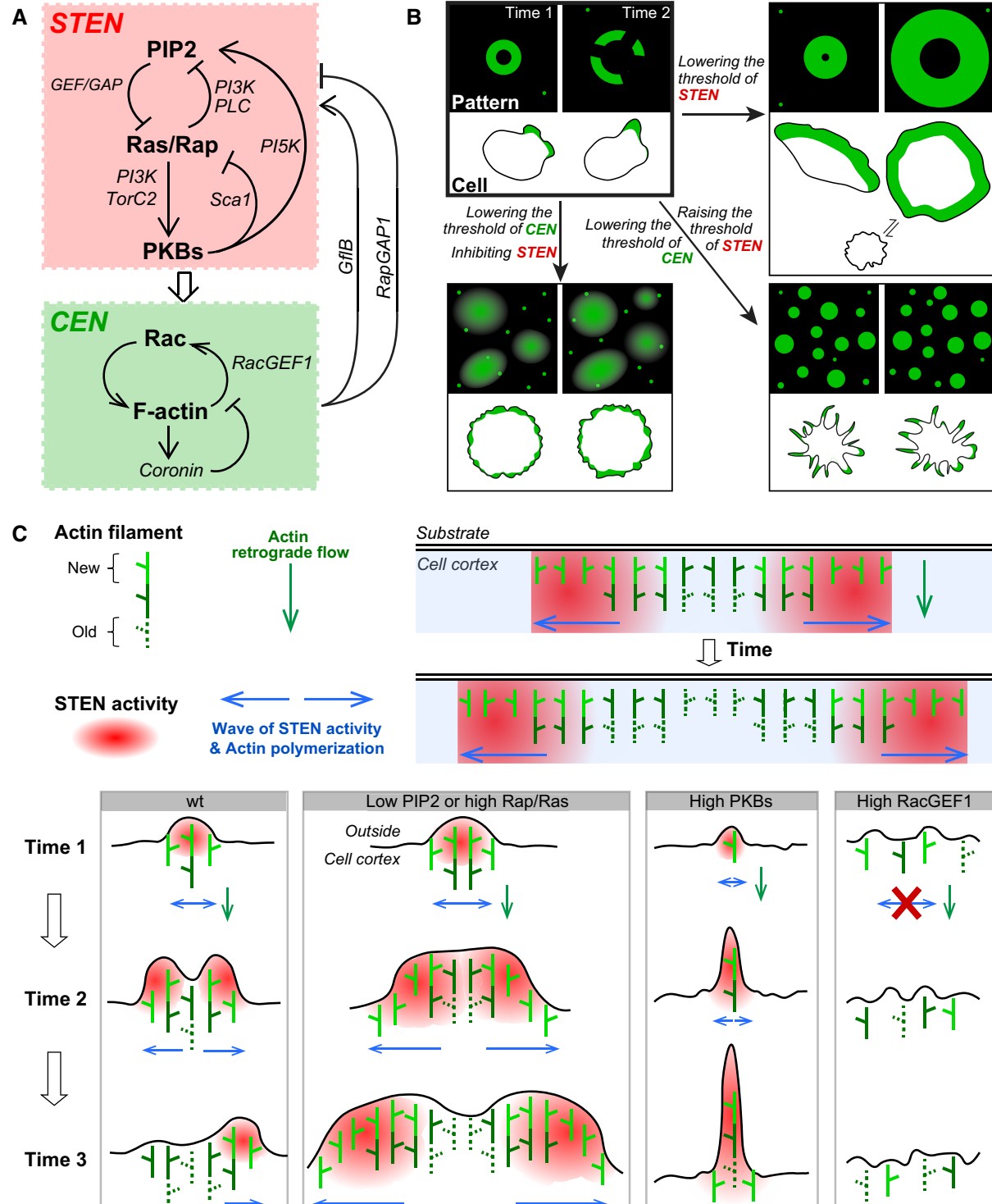

**Figure 9. Models of what molecular feedbacks generate cortical patterns and how wave patterns bring about diverse protrusions.**

A  Summary of molecular interconnections underlying cortical wave self-organization.

B  Diagram of different profiles of cortical patterns and corresponding cell morphology.

C  Upper right illustrates a simplified case where the wave propagates along cell cortex without inducing membrane deformation such as waves at the basal surface of an adherent cell. Individual actin filaments do not dislocate laterally, only elongate perpendicularly to the cell membrane generating retrograde flow (green arrow). STEN activity propagates along the cortex as waves (blue arrows) and leads to CEN activation promoting new actin polymerization (bright green). The successive polymerization events among filaments thus spread along the cortex in the form of waves (also blue arrows). After STEN moves away, actin polymerization ceases, and existing filaments gradually depolymerize. Lower panels show that waves with different properties result in different types of protrusions.

Our results suggest that the speed and range of the traveling pulse depend on an intricate balance between the strengths of the various loops and the resulting threshold (Fig 3). Theoretical studies have long proposed that the level of inhibition in the medium acts as a "controller species" (Fife, 1984; Tyson & Keener, 1988) regulating wave speed. Modulating excitability through optical stimulations to alter wave characteristics has been demonstrated in the Belousov–Zhabotinsky reaction (Sakurai *et al*, 2002). In the mammalian cortex, use of electric fields to alter neural threshold has been shown to change wave speed (Richardson *et al*, 2005). Here, we demonstrate that this theoretical relationship between wave speed and threshold holds in the cellular signaling system.

The system studied has allowed us to also probe the range of traveling waves. Unlike action potentials that propagate along axons indefinitely with little damping, the biochemical cortical waves had finite range. The theory indicates that wave range is correlated with threshold. Wave range becomes pertinent in migrating single cells as the size of the wave dictates the morphology of the protrusion that drives the cell. In wild-type cells, the cortical waves stopped after traversing a short distance on the membrane—which translated to a finite protrusion size in single cells. Lowering the threshold of the network expanded the size of these protrusions ultimately to envelop the whole cortex—leading to fan-shaped and oscillatory cells (Miao *et al*, 2017).

### A network view of diverse cellular protrusions

We found that perturbing the networks led to different cortical patterns of F-actin, and the same perturbations altered cellular protrusions correspondingly (Fig 9B). In unperturbed cells, the waves have a finite range, and the protrusions appear as macropinosomes and pseudopodia with a defined size. Lowering the threshold of STEN increases the speed and range of waves, and cells generate much expanded, sheet-like protrusions resembling lamellipodia. This promoted cell migratory mode transitions, as we showed previously (Miao *et al*, 2017). On the other hand, raising the threshold of STEN restricts the outward propagation of patterns, and cells make spiky, filopodia-like protrusions. When STEN is inhibited due to feedback from elevated CEN, actin polymerization displays diffuse patches and puncta. Correspondingly, without STEN, cells are not capable to form sustained protrusions and show ruffle-like protrusions instead.

Specific, individual molecular regulators of various protrusions such as macropinosomes, lamellipodia, filopodia, and ruffles have been proposed in the literature (Mejillano *et al*, 2004; Mattila & Lappalainen, 2008; Mahankali *et al*, 2011; Krause & Gautreau, 2014; Veltman *et al*, 2016). Our findings suggest that these regulators are all feeding into the same overall molecular machinery. Here, we rapidly switched among these protrusions and traced each transition to a change in the behavior of the networks. Thus, we propose that global properties of the Ras/Rap-centered STEN and Rac/F-actin-centered CEN are the actual determinants controlling the entire spectrum of protrusions seen in cell migration.

### The coupled STEN-CEN model explains how wave patterns control protrusions

We propose that wave patterns generated by the coupled excitable networks serve as a higher order organizer of cellular morphology

(Fig 9C). Diverse protrusions differ by how far they extend outwards from the cell cortex and how wide they expand laterally. Actin filament elongation against the cell membrane generates protrusive forces, while their parallel organization defines the lateral range of protrusions. Thus, a slowly moving organizer would lead to narrower and longer projections, while a rapidly moving one would produce wider and shallower protrusions.

In our proposal, the STEN-CEN waves laterally coordinate successive and transient actin polymerization events along the cell cortex (Fig 9C). This lateral coordination is required to produce meaningful cellular structures whose scale largely exceeds that of individual actin filaments. Simple diffusion by organizing molecules would be insufficient (Deneke & Di Talia, 2018). Instead, the reaction–diffusion waves by the STEN-CEN system fulfill this lateral communication. As the wave propagates laterally along the cortex, it initiates transverse actin polymerization events which showed no dislocation (Fig 2B). On the other hand, the duration of STEN activity controls the extent of outward polymerization at a given spot, thus also influencing protrusive properties.

Innately, waves with a limited range generate cup-like macropinosomes with defined sizes due to wave stopping. Asymmetric wave propagation can lead to pseudopodia-like protrusions. When the speed and range of waves increase—for example, after lowering PIP2 or raising Rap/Ras activities—parallel polymerization events sweep laterally more quickly generating widened protrusions resembling lamellipodia. When the STEN waves propagate very slowly, for example, after PKBA recruitment, local CEN receives constant input from STEN and continuously stimulates actin polymerization, leading to long, thin protrusions. When STEN waves are inhibited after RacGEF1$^{\Delta N}$ recruitment, polymerization cannot be organized or sustained, promoting only ruffles. Thus, STEN-CEN waves with different speed and range can lead to numerous collective patterns of polymerizing filaments in time and space, creating protrusions of diverse properties.

## Materials and Methods

### Cells and plasmids

Wild-type *Dictyostelium discoideum* cells of the AX2 strain, obtained from the R. Kay laboratory (MRC Laboratory of Molecular Biology, UK), were used in this study. Cells were used within 2 months of thawing from frozen stocks. To improve the efficiency of co-expressing multiple proteins, heat-killed *Klebsiella aerogenes* (KA) was supplemented to all cell culture dishes.

Plasmids encoding the CID system are mostly described previously (Miao *et al*, 2017). In addition, to make mCherry-FRB-RacGEF1$^{\Delta N}$ (pCV5), sequence encoding 611–1,218 amino acids of RacGEF1 were amplified from genome DNA and inserted into the plasmid mCherry-FRB-MCS, with five repeats of AGTGCTGGTGGT between FRB and RacGEF1$^{\Delta N}$. Constructs based on pCV5 and pB18 are G418 resistant, and pDM358 is hygromycin resistant. Different pCV5 or pB18 constructs can be used together during cell transformation to co-express several proteins, while the cells are selected using only G418. Here are some key plasmid combinations used (plasmids are named based on composition of fusion proteins from N to C terminus):

Recruiting Inp54p, LimE as readout: mCherry-FRB-Inp54p (pB18); myr-FKBP-FKBP/LimE-YFP (pDM358).

Recruiting Inp54p, $PH_{crac}$ as readout: mCherry-FRB-Inp54p (pB18); myr-FKBP-FKBP/$PH_{crac}$-YFP (pDM358).

Recruiting PKBA, LimE as readout: PKBA-mCherry-FRB (pCV5); myr-FKBP-FKBP/LimE-YFP (pDM358).

Recruiting PKBA, LimE and $PH_{crac}$ as readout: PKBA-CFP-FRB (pCV5); myr-FKBP-FKBP (pCV5); $PH_{crac}$-YFP/LimE-RFP (pDM358).

Recruiting RacGEF1$^{\Delta N}$, LimE as readout: mCherry-FRB-RacGEF1$^{\Delta N}$ (pCV5); myr-FKBP-FKBP/LimE-YFP (pDM358).

## Cell fusion

Growth-phase cells grown in suspension or on cell culture dishes were harvested, washed, and resuspended in SB (17 mM Soerensen buffer containing 15 mM $KH_2PO_4$ and 2 mM $Na_2HPO_4$, pH 6.0) at a density of $1.5 \times 10^7$ cells/ml. 7–10 ml of cells was put into a 50-ml conical tube and rolled gently for ~ 30 min, to promote visible cell clusters formation. 800 μl of rolled cells was transferred to a 4-mm-gap electroporation cuvette, using pipette tips whose edges were cut off to avoid breaking clusters. Electroporation was carried out with the following settings: 1,000 V, 3 μF for once, and 1,000 V, 1 μF for three times, with 1–2 s between pulses. Then, 50 μl of cells was transferred to the center of a well in an 8-well chamber and was left still for 5 min. 450 μl of SB plus 2 mM $CaCl_2$ and 2 mM $MgCl_2$ was added to the well and was pipetted briefly to suspend the cells evenly. After enough cells settled down to the bottom, all media together with excess floating cells were removed and 450 μl of new SB plus 2 mM $CaCl_2$ and 2 mM $MgCl_2$ was gently added to the well against the wall. Cells were left still for 1 h for recovery before imaging experiments.

## Microscopy

Zeiss LSM780 and Zeiss LSM800 GaAsP single-point laser-scanning microscope were used for confocal image acquisition, where bottom focal planes were focused on to capture basal cortical waves.

## Biosensors peak distance

Peak distance was computed using a custom-written MATLAB script. Intensities of the red and green channels in a manually selected region were filtered using a moving average filter of length 19 pixels. Based on these filtered profiles, the locations of the maximum intensity for each color were obtained using the max function. The distance between the two maxima was then used.

## Wave speed analysis

To measure wave speed, a custom MATLAB script was used. Briefly, wave fronts were segmented at subsequent frames of videos. Based on these segmentations, the number of patches and mean area was computed. To measure mean speed, the distance from each pixel on the boundary of a wave front in frame $n + 1$ to closest edge of a wave in frame $n$ was computed. This number was compared to low and high thresholds (0.1 and 5 mm, respectively) to ensure that it represented movement of the same wave front.

Data were then averaged over all fronts in a frame to record average wave speed. The "fraction of fastest pixels" is the fraction of all pixels at the edge of a wave front whose frame-to-frame displacement is at least one standard deviation higher than the mean, where the mean and standard deviation are computed for all frames in the video. Data from 30 frames at 15-s intervals were used to compute averages before CID recruitment. Thereafter, 30 frames were allowed to elapse to allow CID to have effect, before data for a further 30 frames were used to compute the post-recruitment numbers.

## Mathematical modeling and analysis

### Coupled excitable system wave simulations and perturbations

The excitable waves were modeled through reaction–diffusion equations. The STEN was set up as an activator ($F_S$)–inhibitor ($R_S$) system as shown below (Bhattacharya & Iglesias, 2018b):

$$\frac{\partial F_S}{\partial t} = D_{F_s}\nabla^2 F_S - (a_{1s} + a_{2s}R_S)F_S + \frac{a_{3s}F_S^2}{a_{4s}^2 + F_S^2} + a_{5s} + U_N$$

$$\frac{\partial R_S}{\partial t} = D_{R_s}\nabla^2 R_S + \in_S(-R_S + c_{1s}F_S)$$

The non-linear term in the activator equation contributes to the positive feedback, while the epsilon (ε) in the inhibitor equation accounts for the slowness in the response of the inhibitor. When the activator receives a supra-threshold input, the autocatalytic feedback leads to a sharp rise in activity, creating the wave front. The inhibitor, albeit slowly, accumulates to ultimately subdue the activator concentration—creating the wave back. This change in concentrations is better visualized through trajectories in phase space. A large delay in inhibitor response causes the initial trajectory of the state to be nearly horizontal as shown in Fig 2D. After the inhibitor subdues the activator response, it then decays back to resting concentration, which is reflected as the vertical descent of the state toward the final equilibrium. This decay time of the inhibitor creates the refractory period. Coupled with diffusion across adjacent excitable elements, this results in the propagation of a wave "pulse", identified by a sharp rise and fall in activity.

The threshold of the excitable system is related to the difference between the initial inhibition level ($R_0$) and the minimum of the cubic nullcline (Fig 2D). We used two parameters to modulate this threshold: one, the strength of the autocatalytic positive feedback (parameter $a_{3s}$), which alters the cubic nullcline shape as in Appendix Fig S3I (Inp54p, red activator nullcline), and two, the slope of the inhibitor nullcline (parameter $c_{1s}$), which changes the initial inhibition level $R_0$, as in Appendix Fig S3J (PKBA, red inhibitor nullcline). Each of these perturbations was simulated by linearly altering the target parameter gradually over a period of 10 simulation time units. The term, $U_N$, represents the stochastic input to STEN. This was modeled as a zero-mean Gaussian white noise process with a standard deviation given as: $\sigma_N = U_b + Z_c - W_c$. This consisted of three parts: one, a basal level of stochastic fluctuations that triggers the cell even in the absence of the cytoskeleton ($U_b$); two, a positive feedback from the cytoskeletal excitable network ($Z_C$); and three, a negative feedback from CEN ($W_C$). The

feedback from CEN was incorporated in the stochastic term so as to account for the rapidly changing local dynamics of cytoskeletal activity in contrast to the uniform long-ranging effects of altering direct feedback connections.

CEN was modeled similar to STEN, as an activator ($F_C$)–inhibitor ($R_C$) system with slight alterations to the parameter values.

$$\frac{\partial F_C}{\partial \mathrm{t}} = D_{F_c}\nabla^2 F_C - (a_{1c} + a_{2c}(R_C - s_{\mathrm{in}}))F_C + \frac{a_{3c}F_C{}^2}{a_{4c}^2 + F_C{}^2} + a_{5c}$$

$$\frac{\partial R_C}{\partial \mathrm{t}} = D_{R_c}\nabla^2 R_C + \varepsilon_C(-R_C + c_{1c}F_C)$$

In contrast to the bistable oscillatory network proposed earlier in the literature (Huang *et al*, 2013), CEN was made excitable in these simulations. This was done to recreate the distinct puncta observed in the trailing band of the propagating wave. Compared to STEN, the major difference in CEN was in the epsilon term, which was significantly larger than STEN, causing the CEN state trajectory to be shorter in phase space as illustrated in Fig 2E. This also ensured that CEN activity was contained in space and did not spread in the form of a wave. The threshold of CEN was adjusted through three parameters: one, the strength of the autocatalytic feedback loop (parameter $a_{3c}$) as in Appendix Fig S3J (PKBA, green activator nullcline); two, the slope of the inhibitor nullcline (parameter $c_{1c}$) as in Appendix Fig S3K (RacGEF1, green inhibitor nullcline); and three, the self-degradation term of the CEN activator (parameter $a_{1c}$), as in Appendix Fig S3K (RacGEF1, green activator nullcline). A sufficiently large stochastic input was added to CEN so that all points oscillated randomly as reported earlier in the literature (Huang *et al*, 2013). The coupling from STEN to CEN was incorporated through the $s_{\mathrm{in}}$ term, where $s_{\mathrm{in}} = s_c(R_S - R_0)$. An input from STEN ($R_S$) causes the CEN activator nullcline to be raised vertically as shown in Fig 2F, dramatically reducing threshold and forcing the CEN trajectory to follow the STEN input (Fig 2G—left). If the decay of the STEN input was drastically slowed (Appendix Fig S2A), the subsequent CEN firings occurred at this high basal level.

On the down-stroke of the STEN input, the CEN nullcline starts to descend along with the CEN state. This creates a brief refractory period (Fig 2G—middle). If the down-stroke is fast enough (Appendix Fig S2B), the refractory period is sustained and CEN cannot fire again. For an intermediate down-stroke (Fig 2F), soon the state catches up to the nullcline and fires stochastically creating a second peak (Fig 2G—right). Note that this second peak is higher than the basal level of CEN firings (Fig 2F) but lower than the first peak.

The STEN-CEN firings translated to propagating waves in space because of the diffusion terms in the above equations. The output readouts were taken from the inhibitors of both modules, scaled and normalized uniformly. Diffusion was simulated through discretizing the simulation space using the central-difference approximation. All simulations were carried out using the SDE toolbox of MATLAB (Picchini, 2007). The simulations introducing wave characteristics in Fig 3 were carried out on a finer 1D grid (3,000 × 1) for a proof-of-principle purpose. The rest of the results were simulated on a coarser grid (200 × 200 in 2D). Parameters for each simulation are provided in Appendix Table S1.

### Feedback from CEN to STEN

Feedback from the cytoskeletal to the signaling network consisted of fast, local positive feedback ($Z_C$), and slow, global negative feedback ($W_C$). As previously shown, these complementary loops can explain several aspects of cytoskeleton-dependent cell polarity (Huang *et al*, 2013; Shi *et al*, 2013). This feedback was coupled with a stochastic component to account for cytoskeletal fluctuations. These were set up as follows:

$$\frac{\partial Z_C}{\partial \mathrm{t}} = D_{Z_c}\nabla^2 Z_C - p_1 Z_C + p_2 F_C$$

$$\frac{\partial W_C}{\partial \mathrm{t}} = D_{W_c}\nabla^2 W_C - p_3 W_C + p_4 F_C$$

The input to the feedback module was from CEN activator component ($F_C$). This caused the positive feedback to essentially mimic the CEN activator. The negative feedback, however, remained at a low unless there was long-lived activation of the whole cell—as in the RacGEF1 case (Appendix Fig S3K). To simulate the global nature of the inhibitor, its diffusion constant was chosen to be sufficiently large such that $W_C$ is spatially independent. The strength of the negative feedback in steady-state is assumed to be twice that of the positive feedback.

### Wave characteristics, threshold, and dispersion

The dependence of wave speed on threshold has typically been analyzed using singular perturbation theory, in which case a formula for wave speed is available and valid in the singular limit (i.e., $\varepsilon = 0$; Tyson & Keener, 1988). For the simulations shown in Fig 3, $\varepsilon$ was chosen to be 0.03, and so, the $R_{\mathrm{stop}}$ values obtained analytically and through simulations were similar. For higher values of $\varepsilon$, as long as the system remained excitable, the fact that the wave is faster at a lower threshold was valid (Appendix Fig S3H).

Unlike neuronal excitable waves that propagate with little damping, cellular waves stop after traversing a certain distance on the cortex. The mechanism of wave stopping that we analyzed in Fig 3 is similar to the notion of wave-pinning reported earlier in the literature (Mori *et al*, 2008) for bistable systems. A wave is triggered at a certain inhibition level, and it starts to propagate. For this wave to naturally stop, the level of inhibition must continually rise in space (so as to reach $R_{\mathrm{stop}}$) as the wave propagates by triggering adjacent elements. To achieve this, Mori *et al* suggest that the diffusion coefficient of the inhibitor must be significantly greater than that of the activator. In fact, wave stopping can be reached if the *molecular dispersion* of the inhibitor is greater than that of the activator. Molecular dispersion (the square root of the product of diffusion coefficient and lifetime) can be greater for the inhibitor even for a smaller diffusion coefficient as the inhibitor is inherently longer-lived than the activator (resulting in the refractory period). It is the time delay between the two that allows the activator to spread further initially. Thus, wave stopping can still be achieved for lower diffusion of the inhibitor in space, provided its dynamics is made faster in time.

Overall, for wave stopping, the combination of diffusion and time delay of the inhibitor must be such that a zero wave speed situation is possible (Dockery *et al*, 1988; Dockery & Keener, 1989; Kessler & Levine, 1989). In Fig 3, a large time delay (small $\varepsilon$) is chosen and that necessitates a larger diffusion coefficient for the

inhibitor as compared to the activator. However, even for a lower diffusion of the inhibitor, wave stopping can still be achieved by decreasing the delay—provided the system remains excitable. Thus, the effect of threshold on wave stopping is visible for a range of values for diffusion coefficients and time delays (Appendix Fig S3H) as long as the parameters are such that the inhibitor can catch up to the activator. For both low diffusion coefficient and large time delay of the inhibitor, the wave will spread indefinitely.

*Level set simulations*

To determine the effect of these changes on morphology, we simulated cell behavior using level set methods (LSM), as previously described (Yang *et al*, 2008; Miao *et al*, 2017), where the cell boundary is defined as the zero level set of a signed distance function.

The cell is then subjected to stresses obtained from the wave simulations. Activity obtained from these simulations was converted to forces normal to the membrane. We use a viscoelastic model of the cell to determine the local velocity of the level set. Specifically, we used:

$$\dot{x}_{\mathrm{mem}} = -(K/D)x_{\mathrm{cor}} + (1/D + 1/B)\sigma_{\mathrm{tot}}$$

$$\dot{x}_{\mathrm{cor}} = -(K/D)x_{\mathrm{cor}} + (1/D)\sigma_{\mathrm{tot}}$$

where $\sigma_{\mathrm{tot}}$ is the stress applied on the cell, $x_{\mathrm{mem}}$ and $x_{\mathrm{cor}}$ are the local displacements of the membrane and cortex, respectively, and *K*, *D,* and *B* are viscoelastic components of the cell describing the elasticity (*K*) and viscosity (*D*) of the membrane, and the viscosity of the (*B*) of the cytoplasm. The total stress applied to the cell incorporated the effects of surface tension, volume conservation, and external forces. The values of the parameters used are provided in Appendix Table S1.

**Expanded View** for this article is available online.

## Acknowledgements

The authors would like to thank all members of the Devreotes and Iglesias laboratories as well as members of T. Inoue, M. Iijima, and D. Robinson laboratories (Johns Hopkins University) for helpful suggestions. We sincerely thank A. Kortholt (U of Groningen, the Netherlands), P. Charest (U of Arizona), TJ Jeon (Chosun U, South Korea), and R. Firtel (UCSD) for kindly sharing constructs. This work was supported by NIH grant R35 GM118177 (to P.N.D.), AFOSR MURI FA95501610052, DARPA HR0011-16-C-0139, as well as NIH Grant S10 OD016374 (to S. Kuo of the JHU Microscope Facility).

## Author contributions

YM designed and performed a majority of experiments; SB conducted all computational simulations; TB, BA-S, and YL contributed to experiments; YM, SB, and PAI analyzed the data; and YM, SB, PAI, and PND wrote the manuscript with inputs from TI.

## Conflict of interest

The authors declare that they have no conflict of interest.

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
