## [Review Process File · Molecular Systems Biology]

Wave patterns organize cellular protrusions and control cortical dynamics

Yuchuan Miao, Sayak Bhattacharya, Tatsat Banerjee, Bedri Abubaker-Sharif, Yu Long, Takanari Inoue, Pablo A. Iglesias and Peter N. Devreotes.

Review timeline:

Submission date:	2 nd August 2018
Editorial Decision:	17 th October 2018
Revision received:	21 st December 2018
Editorial Decision:	22 nd January 2019
Revision received:	31 st January 2019
Accepted:	4 th February 2019

Editor: Maria Polychronidou

Transaction Report:

1st Editorial Decision

17th October 2018

Thank you again for submitting your work to Molecular Systems Biology. I would like to apologize once again for the delay in sending you a decision on your manuscript. As I already mentioned in our earlier correspondence, we had initially secured two reviewers but unfortunately one of them never returned a report despite a series of reminders and phone calls. In order to perform an informed evaluation without relying on the single opinion of reviewer #1, we had to invite new reviewers and this considerably delayed the process. Two additional reviewers #3 and #4 accepted to evaluate the study. We have now heard back from reviewer #3, and since their overall recommendation is similar to that of reviewer #1, we have decided to make a decision to not delay the process further. If we received comments from reviewer #4 within the following days, we will forward them to you.

As you will see below, the reviewers think that the study seems to be a valuable contribution to the field. However, they raise a series of concerns, which we would ask you to address in a revision.

Overall, the reviewers think that further experimental and modeling analyses as well as additional controls are required to better support the conclusions of the study. The reviewers make constructive suggestions in this regard. I think their recommendations are clear and there is therefore no need to repeat the comments listed below. Please feel free to contact me in case you would like to discuss in further detail any of the issues raised by the reviewers.

REFeree REPORTS

Reviewer #1:

Miao and Bhattacharya et al., used cortical wave patterns in the Dictyostelium to dissect the

molecular network underlying the dynamic excitable behaviours which are important for cell protrusions and motility. Building upon their previous work, they have now systematically mapped the dynamics of many cortical molecules and grouped them into two networks (signal transduction excitable network, or STEN; the cytoskeleton excitable network, or CEN). They sought to establish a link between the properties of these networks and features of wave propagation, in particular, they were interested in the speed, range or size of the waves. They developed experimental strategy to acutely change individual nodes of the network. They found that changing the PIP2 level increased wave speed and range. Conversely increasing PKB activities caused a decrease in the speed and range of the waves. Activating RacGEF triggers massive actin polymerization but not in the form of propagating waves. They then concluded that PIP2 and PKB participate in the positive and negative feedback loops of STEN, respectively, while Rac is part of the positive feedback in CEN.

Overall I think this paper addresses fundamental questions in cortical excitability that remain largely elusive. It is a very comprehensive study and provided a ton of new information that would likely make the field to digest for a long time. And it is visually compelling and very quantitative. I would be happy to support its publication if some of the comments below can be addressed.

One of the major concepts in the manuscript is that threshold sets the wave propagation speed and their range of the propagation. However, there are some confusions as to how this threshold is set by the components in the excitable system, and ultimately biological systems. In the model (Fig 3), it appears that it is set by the initial inhibitor level in the activator-inhibitor system. Experimentally, it was perturbed by increasing the strength of the positive feedback (interpreted as such), which changes the slope of the nullcline instead. Perhaps this can be clarified by adding some modelling results, or discussed. However, this is an important point. The rest of experiments were interpreted based on the assumption that inhibitor level is the only parameter that could affect wave propagation speed, while theoretically changing inhibitor may be sufficient but not necessary for such outcome. In addition, I would be interested to see how the wave speed in the model depends on the threshold (i.e. whether there is an upper limit, and whether it only applies to certain regimes-- relatively slow waves for example), but this is not required if it adds a considerable amount of new work.

Specific comments:

1. I would like to see some clarifications or quantifications of the two bands for proteins in CEN waves. Do the two peaks share the same asymmetry, i.e. leading edge is sharper, and trailing edge has shallower slope? The quantifications in Fig 1 seem to suggest that they have opposite symmetry (not explicitly mentioned in the text), while the model suggests otherwise. Either line profiles or time profiles could be helpful.
2. It may be easier to the readers to illustrate experimental data corresponding to the different scenarios of Fig 2d/e/f as early as possible (something like the profiles in Fig 1a, Fig 4e, Fig 2a). Upon reading Page 5, one has the impression that CEN waves are double-banded (which was used as a defining feature to differentiate with STEN), but that is not always true (situation in Fig 2e), unless the authors meant to say that only 2f is experimentally observed. To what extent scenario 2d (double band) and 2f (double band but the second one is a punctate trailing band) are interchangeable in the same wave (depending on the time point, it could be either one, see Fig 1a)?
3. Based on Fig 4d and the definitions in Fig 1, PIP2 (inferred from PI5K) may behave in a way that is neither STEN nor CEN, is it?
4. I wonder whether the possibility that reduced PIP2 decreases membrane tension and stimulates excitability has been considered. After all, if waves propagate faster when PIP2 is reduced/depleted, it is hard to imagine that there are still some PIP2 left and these PIP2 (if any) will be chasing after Ras and propagating faster under this condition (it is not entirely impossible but not the most likely). I feel the experimental evidences supporting the mutual inhibition between Ras and PIP2 as the core feedback for STEN is weak (actually, close to non-existent).
5. In the model of CEN, Rc has a local inhibitory effect on Fc, but the negative feedback from CEN to STEN is modeled as global (Wc). Does this imply the requirement of two separate underlying mechanisms?
6. Page 6, "To illustrate the effect of this coupling, three different spatial wave profiles were applied to CEN...". Does this refer to the initial condition and because CEN have low rate of diffusion, these profiles are more or less maintained? I hope this can be elaborated.
7. I have some reservations of interpreting PKB as part of the negative feedback. If that is indeed the case, one presumably would see reduced recruitment of STEN activators.
8. Because PIP2 was also mentioned later in the work, it is better to specify PH domain as PH crac

or PIP3 to minimize confusions.

Reviewer #3:

This paper is a long, very detailed study of the role of both signaling and cytoskeletal dynamics in the organization of waves and concomitant protrusive activity in *Dictyostelium*. I found the work very informative and compelling and hence deserving of (eventual) publication. There are some points, however, that confused me, which I would request be clarified before proceeding.

The idea of putting the feedback from CEN to STEN into the noise term (second equation in the Modeling section, p. 77) is certainly unusual and therefor needs to be discussed in greater detail. I imagine that they did this for the positive feedback to allow for slower wave speeds but more patches in the PKB section - is this correct? Is there any evidence that this is also the form the negative feedback should take and how does this noise totally eliminate STEN activity if CEN is increased via the RacGEF construction (p.11)? Do the authors have some particular mechanistic model in mind for this assumption?

The role of inhibitor diffusion on slowing down waves in excitable systems was studied extensively in the BZ reaction system and some papers from that epoch (Dockery et al, *Physics D* 30, 177 [1998]; Kessler et al, *Physics D* 39, 1 [1989]).

A clear prediction of the theory, as shown for example in Fig. 3G, is the STEN waves should "lead" CEN waves. The experiments in Fig. 1 shows LimE ahead of PH, but this does not necessarily disagree with the above prediction as different components of the STEN wave may be delayed as compared to the fastest component. Now, Fig. 1g implies that Rap is at the lead of the STEN wave, and indeed Fig. 1C shows that it proceeds PH by about 5 μ m. Did the authors ever look at RalGDS together with LimE and verify the above prediction? If yes, one picture of this would be worth including as part of Fig. 1.

Although it is presented in the earlier papers, it would be worthwhile for completeness to include some more details of the protrusion model in the modeling section. In any event, the connection between theory and experimental results are as significantly weaker in this section. Perhaps the authors can offer some guidelines as to how future research could correlate the live-cell wave images directly with cell shape changes so as to further test the proposed correspondence between protrusions and these wave actions.

Reviewer #1:

Miao and Bhattacharya et al., used cortical wave patterns in the Dictyostelium to dissect the molecular network underlying the dynamic excitable behaviours which are important for cell protrusions and motility. Building upon their previous work, they have now systematically mapped the dynamics of many cortical molecules and grouped them into two networks (signal transduction excitable network, or STEN; the cytoskeleton excitable network, or CEN). They sought to establish a link between the properties of these networks and features of wave propagation, in particular, they were interested in the speed, range or size of the waves. They developed experimental strategy to acutely change individual nodes of the network. They found that changing the PIP2 level increased wave speed and range. Conversely increasing PKB activities caused a decrease in the speed and range of the waves. Activating RacGEF triggers massive actin polymerization but not in the form of propagating waves. They then concluded that PIP2 and PKB participate in the positive and negative feedback loops of STEN, respectively, while Rac is part of the positive feedback in CEN.

Overall I think this paper addresses fundamental questions in cortical excitability that remain largely elusive. It is a very comprehensive study and provided a ton of new information that would likely make the field to digest for a long time. And it is visually compelling and very quantitative. I would be happy to support its publication if some of the comments below can be addressed.

Response: We would like to thank the reviewer for his/her comments and support of the research.

One of the major concepts in the manuscript is that threshold sets the wave propagation speed and their range of the propagation. However, there are some confusions as to how this threshold is set by the components in the excitable system, and ultimately biological systems. In the model (Fig 3), it appears that it is set by the initial inhibitor level in the activator-inhibitor system. Experimentally, it was perturbed by increasing the strength of the positive feedback (interpreted as such), which changes the slope of the nullcline instead. Perhaps this can be clarified by adding some modelling results, or discussed.

However, this is an important point. The rest of experiments were interpreted based on the assumption that inhibitor level is the only parameter that could affect wave propagation speed, while theoretically changing inhibitor may be sufficient but not necessary for such outcome.

Response: The reviewer is correct in that there are many ways of altering threshold. For example, in the original Figure 3 wave speed increased by decreasing negative feedback. We have now added Appendix Figures S3E and S3F, showing that positive feedback can also be increased to achieve similar results. The main difference is that the positive feedback strength is more sensitive. When we sought to recreate experiments that we interpret as having increased positive feedback (Figure 4), we raised the activator nullcline in the model to lower threshold (Appendix Figure S3I). A paragraph has been added in the main text (page 8, paragraph 3) discussing this point.

In addition, I would be interested to see how the wave speed in the model depends on the threshold (i.e. whether there is an upper limit, and whether it only applies to certain regimes - relatively slow waves for example), but this is not required if it adds a considerate amount of new work.

Response: We have added Appendix Figure S3G studying the effect of a threshold range on wave speed. We show that higher thresholds decrease wave speeds until no appreciable wave spread occurs. Lower thresholds increase wave speeds until synchronized oscillations start – at which point wave speed is not meaningful. A line has been added in the main text explaining this point (page 8, paragraph 4).

Specific comments:

1. *I would like to see some clarifications or quantifications of the two bands for proteins in CEN waves. Do the two peaks share the same asymmetry, i.e. leading edge is sharper, and trailing edge has shallower slope? The quantifications in Fig 1 seem to suggest that they have opposite symmetry (not explicated mentioned in the text), while the model suggests otherwise. Either line profiles or time profiles could be helpful.*

Response: The reviewer raises a good point here. The discrepancy was due to the different ways of calculating the profiles. Experimentally, we had used a rectangle to capture the profile. In the theoretical curves, we used a line. The former has the effect of smoothing the resultant data as it averaged intensity across several pixels. In contrast, the single line was quite sensitive to different points. For consistency, we have now also used a rectangle to compute the activity profile of the simulation data (Figure 2H) which now shows an asymmetry similar to that of the experiments – with a sharp leading edge and a shallower trailing edge.

2. *It may be easier to the readers to illustrate experimental data corresponding to the different scenarios of Fig 2d/e/f as early as possible (something like the profiles in Fig 1a, Fig 4e, Fig 2a). Upon reading Page 5, one has the impression that CEN waves are*

double-banded (which was used as a defining feature to differentiate with STEN), but that is not always true (situation in Fig 2e), unless the authors meant to say that only 2f is experimentally observed. To what extent scenario 2d (double band) and 2f (double band but the second one is a punctate trailing band) are interchangeable in the same wave (depending on the time point, it could be either one, see Fig 1a)?

Response: We apologize for the confusion. The original Figures 2D and 2E were to illustrate the functioning of the coupled model, showing how different profiles in STEN led to corresponding patterns in CEN. In fact, only Figure 2F (now 2H) had a STEN pattern that was experimentally observed. To avoid confusion, the original panels 2D and 2E have been moved to Appendix Figure S3A and S3B.

3. *Based on Fig 4d and the definitions in Fig 1, PIP2 (inferred from PI5K) may behave in a way that is neither STEN nor CEN, is it?*

Response: The PIP2 pattern (now shown in Figure 4K) behaves as the reciprocal of other STEN components – i.e. decreasing when other STEN components increase. It mimics the ‘back’ state in which positive feedback is achieved through complementary negative regulation (Figure 4L). This was illustrated in detail in our previous work (Miao et al. 2017; Li et al. 2018). In the simulations for this paper we have used a simplified model that does not explicitly break down this reciprocal regulation and relied on a single positive feedback.

4. *I wonder whether the possibility that reduced PIP2 decreases membrane tension and stimulates excitability has been considered. After all, if waves propagate faster when PIP2 is reduced/depleted, it is hard to imagine that there are still some PIP2 left and these PIP2 (if any) will be chasing after Ras and propagating faster under this condition (it is not entirely impossible but not the most likely). I feel the experimental evidences supporting the mutual inhibition between Ras and PIP2 as the core feedback for STEN is weak (actually, close to non-existent).*

Response: As suggested by the reviewer, we have carried out further experiments in Latrunculin-treated cells in which the cytoskeletal function was inhibited. As shown in the new Figure 4J and Movie EV5, wave propagation was enhanced after reducing PIP2, suggesting that PIP2 directly regulates STEN activity, independent of membrane tension. The reviewer’s concern about low levels of PIP2 has been previously addressed. In fact, biosensors for PIP2 show that a zone of reduced/depleted PIP2 propagates along with the wave (Gerisch et al. 2011; Miao et al. 2017). Our results that lowering PIP2 and increasing Ras/Rap activity have similar results suggest that there is mutual inhibition between Ras/Rap and PIP2. As we point out in discussion, there are likely other redundant components in these feedback loops that can sustain wave propagation even with low levels of PIP2.

5. *In the model of CEN, Rc has a local inhibitory effect on Fc, but the negative feedback from CEN to STEN is modeled as global (Wc). Does this imply the requirement of two separate underlying mechanisms?*

Response: Yes, two separate underlying mechanisms are needed. Whereas R_c acts as the inhibitor to the excitable network of CEN, W_c is part of the feedback from CEN to STEN (Figure 4A).

6. *Page 6, "To illustrate the effect of this coupling, three different spatial wave profiles were applied to CEN...". Does this refer to the initial condition and because CEN have low rate of diffusion, these profiles are more or less maintained? I hope this can be elaborated.*

Response: The three different spatial and temporal STEN profiles were used to illustrate the working of the STEN-CEN coupling. For example, the step input (Appendix Figure S3A) was achieved by drastically slowing the decay of the STEN profile, while the pulse input (Appendix Figure S3B) was achieved by increasing it. For the profile shown in Figure 2H, the output is taken directly from the signaling excitable network. The details are included in the Methods section (page 21, paragraph 1).

7. *I have some reservations of interpreting PKB as part of the negative feedback. If that is indeed the case, one presumably would see reduced recruitment of STEN activators.*

Response: The reviewer is correct, but must note that PKBA is also coupling to CEN, which increases rate of STEN triggering through positive feedback. To isolate the negative feedback role of PKB, we have added experiments in Latrunculin-treated cells where, in absence of CEN, recruiting PKBA inhibits other STEN biosensors (Figure 5G, Appendix Figure S5C, and Movie EV8).

8. *Because PIP2 was also mentioned later in the work, it is better to specify PH domain as PH_{crac} or PIP3 to minimize confusions.*

Response: We now refer to this PH domain as PH_{crac}.

Reviewer #3:

This paper is a long, very detailed study of the role of both signaling and cytoskeletal dynamics in the organization of waves and concomitant protrusive activity in Dictyostelium. I found the work very informative and compelling and hence deserving of (eventual) publication. There are some points, however, that confused me, which I would request be clarified before proceeding.

The idea of putting the feedback from CEN to STEN into the noise term (second equation in the Modeling section, p. 77) is certainly unusual and therefor needs to be discussed in greater detail. I imagine that they did this for the positive feedback to allow for slower wave speeds but more patches in the PKB section - is this correct?

Response: Yes, the reviewer is correct. Incorporating CEN feedback into the STEN noise term has the advantage of increasing STEN triggers locally without affecting wave speed uniformly. This type of feedback was also used in our previous work (Huang et al. 2013). This has been explained in in the methods section (page 20, paragraph 2).

Is there any evidence that this is also the form the negative feedback should take and how does this noise totally eliminate STEN activity if CEN is increased via the RacGEF construction (p.11)? Do the authors have some particular mechanistic model in mind for this assumption?

Response: The negative feedback was also incorporated into the STEN noise term. This was not necessary but was done for consistency. To eliminate STEN activity, the steady-state of the negative feedback was set to be more powerful than that of the positive feedback. The slow nature of the negative feedback also serves to filter out the fast, positive feedback dynamics. We do not have a particular mechanistic model for this assumption, though one possible way that noise levels can increase is by simultaneously increasing “on” and “off” rates of a reaction. This leaves the mean steady state intact, but has the effect of increasing the size of the resultant fluctuations. This information has been added to the methods section (page 22, paragraph 1).

The role of inhibitor diffusion on slowing down waves in excitable systems was studied extensively in the BZ reaction system and some papers from that epoch (Dockery et al, Physics D 30, 177 [1998]; Kessler et al, Physics D 39, 1 [1989]).

Response: We thank the reviewer for these references, which we have now added.

A clear prediction of the theory, as shown for example in Fig. 3G, is the STEN waves should "lead" CEN waves. The experiments in Fig. 1 shows LimE ahead of PH, but this does not necessarily disagree with the above prediction as different components of the STEN wave may be delayed as compared to the fastest component. Now, Fig. 1g implies that Rap is at the

lead of the STEN wave, and indeed Fig. 1C shows that it proceeds PH by about 5 μ m. Did the authors ever look at RalGDS together with LimE and verify the above prediction? If yes, one picture of this would be worth including as part of Fig. 1.

Response: We did the experiment suggested by the reviewer; Appendix Figure S1F shows that RalGDS and LimE peak at similar times. However, we would like to emphasize that the fact that CEN peaks before STEN is not indicative that CEN triggers STEN, merely that CEN has faster kinetics. In our simulations, depending on the precise temporal profile of STEN, it is also possible for CEN to peak before STEN, although we know from our model that STEN is the driver in these cases (Appendix Figure S3C,D). We have added this explanation in the main text (page 7, paragraph 1).

Although it is presented in the earlier papers, it would be worthwhile for completeness to include some more details of the protrusion model in the modeling section. In any event, the connection between theory and experimental results are as significantly weaker in this section. Perhaps the authors can offer some guidelines as to how future research could correlate the live-cell wave images directly with cell shape changes so as to further test the proposed correspondence between protrusions and these wave actions.

Response. This is a great suggestion by the reviewer. We have added details of the level set methods used for protrusion modeling in the methods section. We have also added a new discussion section (page 17, paragraph 1).along with a model figure (Figure 9C) that elaborates our proposed connection between wave patterns and protrusion morphology, suggesting potential mechanisms to test this theory.

Reviewer #4

Miao et al. provide a new installment of the dictyostelium motility saga. Interestingly, they now explain the "PIP3 lake in the actin corral" pattern as a triple wave in which actin subsystem fires twice, once ahead of the Ras subsystem and second time at the wake of the Ras wave. Devreotes and Iglesias had been working on the problem for over a decade and much had been done by them and others in this fairly mature field. Disappointingly, I find that despite all the rich molecular knowledge accumulated, their models still remain heuristic and the molecules are sort of back-projected onto the model modules at the end.

Major comments:

- 1. The idea that the overall observed dynamics is attributable to a couple of excitable networks is fruitful but not new. Van Haastert and Kortholt proposed this idea in 2016 and more is on the way from Ueda et al. (preprint on bioRxiv). The weakest part of the proposed model is the coupling between the two modules called here STEN and CEN. The authors must explain why van Haastert's model is not sufficient and why their coupling is better than the one proposed by van Haastert and colleagues.*

Response: In fact, we proposed a coupling between signal transduction and cytoskeletal networks in our 2013 Nature Cell Biology paper (Huang et al. 2013). However, neither our earlier version, nor the Van Haastert version – in which Ras couples to Actin – provided comprehensive molecular details underlying the feedback connections. Here, first of all, we have expanded the repertoire of molecules in each of the networks. Secondly, we have perturbed “downstream” molecules and altered “upstream” events thus delineating the molecular details of the feedback loops involved. Thirdly, though our proposed theoretical model (Figure 4A) is simpler than that of Van Haastert, it can account for features of the system such as wave organization, wave stopping and its relation to system threshold, characteristics that Van Haastert’s model did not explore. Lastly, our theoretical model can explain and predict the effects of the experimental perturbations.

- 2. While the experiment seems to indicate that the actin subsystem fires at the wake of the Ras wave, the model behavior is far less convincing. Figure 2f shows strong second maximum of actin but the model snapshot suggests that this is the averaging of small sparks induced in the model by strong noise on the refractory descent of the trajectory to the steady state. That such a forced model behavior explains the second maximum visible on Figure 1 is not clear.*

Response: Many experimental figures (such as Figure 1D and Appendix Figure S1A), in fact, suggest that the trailing band is composed of puncta (or sparks). In the model, with sufficient amount of puncta the trailing band can appear continuous as well (Appendix Figure S3D).

Secondly, in the model, the actin subsystem (CEN) fires just after the Ras (STEN) (see Figure 2), while in the experiment, Figure 1, first maximum of CEN is leading STEN and is well

separated in time. Overall, despite repeated statements that the model reproduces experiment extremely well, the comparison of the figures shows the opposite conclusion.

Response: We addressed this question for Reviewer #3. This answer is repeated here for convenience. Appendix Figure S1F shows RalGDS and LimE peak at similar times. However, we would like to emphasize that the fact that CEN peaks before STEN is not indicative that CEN triggers STEN. It just means that CEN has faster kinetics. In our simulations, depending on the precise temporal profile of STEN it is also possible for CEN to peak before STEN, although STEN is the driver in all cases (Appendix Figure S3C,D). We have added this explanation in the main text (page 7, paragraph 1).

3. *The authors spend at least couple of pages on the phenomenon of wave-stopping (pages 7-8). What they demonstrate in the model is the induction of sub-threshold waves, that collapse shortly after they start propagating. This is a well-known phenomenon in excitable dynamics and it is not clear why the authors give it so much of attention. The biological (over)interpretation of this phenomenon should be removed.*

Response: First of all, these are not sub-threshold waves. All of our excitable waves are supra-threshold, as adjacent elements have similar high amplitudes (Figure 2D,E) until the wave stops. The waves that collapse shortly after initiation are simply supra-threshold waves that stop abruptly due to high threshold.

Secondly, to the best of our knowledge, ours is the first study to point out that cortical waves have a finite range and that the range can have drastic effects on protrusive phenotypes. So, we feel that our theoretical emphasis on this phenomenon is justified. In fact, our theoretical study can explain how altering threshold can change wave characteristics and thus predict and explain the experimental perturbations that are shown in this paper.

4. *Page 5 "Annihilation events when two wave bands meet (Supplementary Fig. 1a) demonstrate the excitable nature of the cortical waves." - This nonsense was once published in a popular review but is completely incorrect. All nonlinear chemical waves annihilate upon collision and this annihilation is not an argument in favor of excitable nature. For example, oscillatory waves will also do the same.*

Response: We have changed the word “demonstrate” to “suggest”. Annihilation is however consistent with the excitable nature of the cortical waves. In fact, annihilation is usually one of the “distinguishing qualitative features that characterize excitable waves” (Argentina, Couillet, and Mahadevan 1997). In previous work both from our lab and others we have verified that the cellular cortex is indeed excitable (Nishikawa et al. 2014; Huang et al. 2013; Miao et al. 2017).

Minor comments:

1. *Summary and Introduction, pages 2 and 4. "these structures fall on a continuum", etc. Sloppy language, this is not clear and will not be understandable to the broad readership who do not come from quantitative sciences. Please re-write.*

Response: The phrase “continuum” has been changed.

2. *Page 9 "We deployed PKBA into the CID system" - this is also on many other pages. Again some technical slang here. What do you mean you "deployed" it "into the such and such system"? If this is experiment you describe, you cannot know precisely into which subsystem your perturbation is "deployed". Please revise language.*

Response: Changed, as suggested.

3. *Same page, "the second LimE peak became more diffusive..." here and in many different places, poor English, should be "diffuse", not "diffusive". E.g., page 11 top, "diffused patches".*

Response: Changed, as suggested.

REFERENCES

- Argentina, M., P. Coulet, and L. Mahadevan. 1997. "Colliding Waves in a Model Excitable Medium: Preservation, Annihilation, and Bifurcation." *Physical Review Letters* 79 (15): 2803–6.
- Gerisch, Günther, Mary Ecke, Dirk Wischnewski, and Britta Schroth-Diez. 2011. "Different Modes of State Transitions Determine Pattern in the Phosphatidylinositide-Actin System." *BMC Cell Biology* 12 (October): 42.
- Huang, Chuan-Hsiang, Ming Tang, Changji Shi, Pablo A. Iglesias, and Peter N. Devreotes. 2013. "An Excitable Signal Integrator Couples to an Idling Cytoskeletal Oscillator to Drive Cell Migration." *Nature Cell Biology* 15 (11): 1307–16.
- Li, Xiaoguang, Marc Edwards, Kristen F. Swaney, Nilmani Singh, Sayak Bhattacharya, Jane Borleis, Yu Long, Pablo A. Iglesias, Jie Chen, and Peter N. Devreotes. 2018. "Mutually Inhibitory Ras-PI(3,4)P2 Feedback Loops Mediate Cell Migration." *Proceedings of the National Academy of Sciences of the United States of America* 115 (39): E9125–34.
- Miao, Yuchuan, Sayak Bhattacharya, Marc Edwards, Huaqing Cai, Takanari Inoue, Pablo A. Iglesias, and Peter N. Devreotes. 2017. "Altering the Threshold of an Excitable Signal Transduction Network Changes Cell Migratory Modes." *Nature Cell Biology* 19 (4): 329–40.
- Nishikawa, Masatoshi, Marcel Hörning, Masahiro Ueda, and Tatsuo Shibata. 2014. "Excitable Signal Transduction Induces Both Spontaneous and Directional Cell Asymmetries in the Phosphatidylinositol Lipid Signaling System for Eukaryotic Chemotaxis." *Biophysical Journal* 106 (3): 723–34.

2nd Editorial Decision

22nd January 2019

Thank you for sending us your revised manuscript. We have now heard back from reviewer #1, who agreed to review your revised study. As you will see below, reviewer #1 is satisfied with the modifications made and thinks that the study is suitable for publication.

Before we formally accept the study for publication, we would ask you to address a few remaining editorial issues listed below.

REFEREE REPORTS

Reviewer #1:

The authors have sufficiently addressed my concerns and I think the current version is greatly improved and suitable for publication.

Corresponding Author Name: Peter N. Devreotes and Pablo A. Iglesias

Manuscript Number: MSB-18-8585